# REVISITING LOCALLY SUPERVISED LEARNING: AN ALTERNATIVE TO END-TO-END TRAINING

**Yulin Wang, Zanlin Ni, Shiji Song, Le Yang & Gao Huang**[*]
Department of Automation, BNRist, Tsinghua University, Beijing, China,
`{wang-yl19, nzl17, yangle15}@mails.tsinghua.edu.cn`
`{shijis, gaohuang}@tsinghua.edu.cn`

## ABSTRACT

Due to the need to store the intermediate activations for back-propagation, end-to-end (E2E) training of deep networks usually suffers from high GPUs memory footprint. This paper aims to address this problem by revisiting the locally supervised learning, where a network is split into gradient-isolated modules and trained with local supervision. We experimentally show that simply training local modules with E2E loss tends to collapse task-relevant information at early layers, and hence hurts the performance of the full model. To avoid this issue, we propose an information propagation (InfoPro) loss, which encourages local modules to preserve as much useful information as possible, while progressively discard task-irrelevant information. As InfoPro loss is difficult to compute in its original form, we derive a feasible upper bound as a surrogate optimization objective, yielding a simple but effective algorithm. In fact, we show that the proposed method boils down to minimizing the combination of a reconstruction loss and a normal cross-entropy/contrastive term. Extensive empirical results on five datasets (i.e., CIFAR, SVHN, STL-10, ImageNet and Cityscapes) validate that InfoPro is capable of achieving competitive performance with less than 40% memory footprint compared to E2E training, while allowing using training data with higher-resolution or larger batch sizes under the same GPU memory constraint. Our method also enables training local modules asynchronously for potential training acceleration. Code is available at: `https://github.com/blackfeather-wang/InfoPro-Pytorch`.

## 1 INTRODUCTION

End-to-end (E2E) back-propagation has become a standard paradigm to train deep networks (Krizhevsky et al., 2012; Simonyan & Zisserman, 2014; Szegedy et al., 2015; He et al., 2016; Huang et al., 2019). Typically, a training loss is computed at the final layer, and then the gradients are propagated backward layer-by-layer to update the weights. Although being effective, this procedure may suffer from memory and computation inefficiencies. First, the entire computational graph as well as the activations of most, if not all, layers need to be stored, resulting in intensive memory consumption. The GPU memory constraint is usually a bottleneck that inhibits the training of state-of-the-art models with high-resolution inputs and sufficient batch sizes, which arises in many realistic scenarios, such as 2D/3D semantic segmentation/object detection in autonomous driving, tissue segmentation in medical imaging and object recognition from remote sensing data. Most existing works address this issue via the gradient checkpointing technique (Chen et al., 2016) or the reversible architecture design (Gomez et al., 2017), while they both come at the cost of significantly increased computation. Second, E2E training is a sequential process that impedes model parallelization (Belilovsky et al., 2020; Löwe et al., 2019), as earlier layers need to wait for their successors for error signals.

As an alternative to E2E training, the locally supervised learning paradigm (Hinton et al., 2006; Bengio et al., 2007; Nøkland & Eidnes, 2019; Belilovsky et al., 2019; 2020) by design enjoys higher memory efficiency and allows for model parallelization. In specific, it divides a deep network into several gradient-isolated modules and trains them separately under local supervision (see Figure 1 (b)). Since back-propagation is performed only within local modules, one does not need to store all

---

[*]Corresponding author.

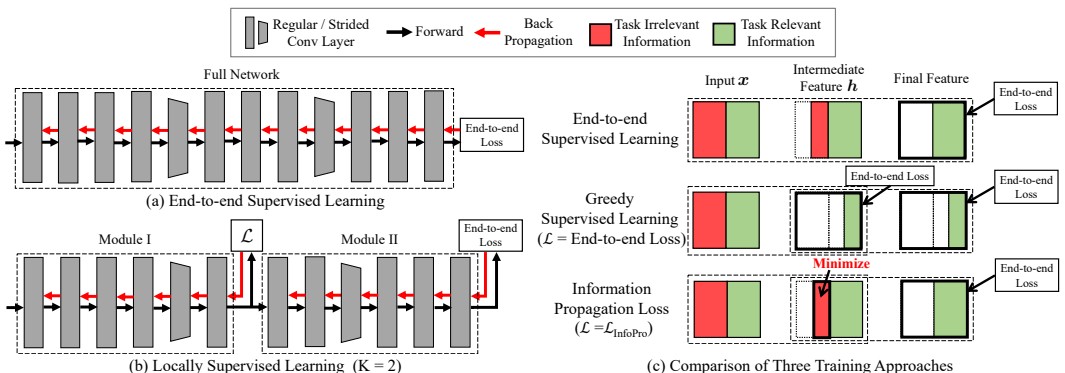

Figure 1: (a) and (b) illustrate the paradigms of end-to-end (E2E) learning and locally supervised learning ($K=2$). "End-to-end Loss" refers to the standard loss function used by E2E training, e.g., softmax cross-entropy loss for classification, etc., while $\mathcal{L}$ denotes the loss function used to train local modules. (c) compares three training approaches in terms of the information captured by features. Greedy supervised learning (greedy SL) tends to collapse some of task-relevant information with the beginning module, leading to inferior final performance. The proposed information propagation (InfoPro) loss, however, alleviates this problem by encouraging local modules to propagate forward all the information from inputs, while maximally discard task-irrelevant information.

intermediate activations at the same time. Consequently, the memory footprint during training is reduced without involving significant computational overhead. Moreover, by removing the demands for obtaining error signals from later layers, different local modules can potentially be trained in parallel. This approach is also considered more biologically plausible, given that brains are highly modular and predominantly learn from local signals (Crick, 1989; Dan & Poo, 2004; Bengio et al., 2015). However, a major drawback of local learning is that they usually lead to inferior performance compared to E2E training (Mostafa et al., 2018; Belilovsky et al., 2019; 2020).

In this paper, we revisit locally supervised training and analyse its drawbacks from the information-theoretic perspective. We find that directly adopting an E2E loss function (i.e., cross-entropy) to train local modules produces more discriminative intermediate features at earlier layers, while it collapses task-relevant information from the inputs and leads to inferior final performance. In other words, local learning tends to be short sighted, and learns features that only benefit local modules, while ignoring the demands of the rest layers. Once task-relevant information is washed out in earlier modules, later layers cannot take full advantage of their capacity to learn more powerful representations.

Based on the above observations, we hypothesize that a less greedy training procedure that preserves more information about the inputs might be a rescue for locally supervised training. Therefore, we propose a less greedy information propagation (InfoPro) loss that aims to encourage local modules to propagate forward as much information from the inputs as possible, while progressively abandon task-irrelevant parts (formulated by an additional random variable named nuisance), as shown in Figure 1 (c). The proposed method differentiates itself from existing algorithms (Nøkland & Eidnes, 2019; Belilovsky et al., 2019; 2020) on that it allows intermediate features to retain a certain amount of information which may hurt the short-term performance, but can potentially be leveraged by later modules. In practice, as the InfoPro loss is difficult to estimate in its exact form, we derive a tractable upper bound, leading to surrogate losses, e.g., cross-entropy loss and contrastive loss.

Empirically, we show that InfoPro loss effectively prevents collapsing task-relevant information at local modules, and yields favorable results on five widely used benchmarks (i.e., CIFAR, SVHN, STL-10, ImageNet and Cityscapes). For instance, it achieves comparable accuracy as E2E training using 40% or less GPU memory, while allows using a 50% larger batch size or a 50% larger input resolution with the same memory constraints. Additionally, our method enables training different local modules asynchronously (even in parallel).

## 2 WHY LOCALLY SUPERVISED LEARNING UNDERPERFORMS E2E TRAINING?

We start by considering a local learning setting where a deep network is split into multiple successively stacked modules, each with the same depth. The inputs are fed forward in an ordinary way, while the gradients are produced at the end of every module and back-propagated until reaching an earlier module. To generate supervision signals, a straightforward solution is to train all the local modules as

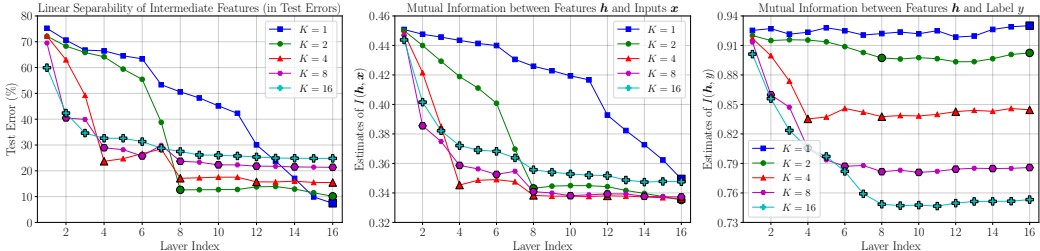

Figure 2: The linear separability (*left*, measured by test errors), mutual information with the input $\boldsymbol{x}$ (*middle*), and mutual information with the label $y$ (*right*) of the intermediate features $\boldsymbol{h}$ from different layers when the greedy supervised learning (greedy SL) algorithm is adopted with $K$ local modules. The ends of local modules are marked using larger markers with black edges. The experiments are conducted on CIFAR-10 with a ResNet-32.

independent networks, e.g., in classification tasks, attaching a classifier to each module, and computing the local classification loss such as cross-entropy. However, such a *greedy* version of the standard supervised learning (greedy SL) algorithm leads to inferior performance of the whole network. For instance, in Table 1, we present the test error of a ResNet-32 (He et al., 2016) on CIFAR-10 (Krizhevsky et al., 2009)

Table 1: Test errors of a ResNet-32 using greedy SL on CIFAR-10. The network is divided into $K$ successive local modules. Each module is trained separately with the softmax cross-entropy loss by appending a global-pool layer followed by a fully-connected layer (see Appendix F for details). "$K=1$" refers to end-to-end (E2E) training.

|  | $K=1$ | $K=2$ | $K=4$ | $K=8$ | $K=16$ |
|---|---|---|---|---|---|
| Test Error | 7.37% | 10.30% | 16.07% | 21.19% | 24.59% |

when it is greedily trained with $K$ modules. One can observe a severe degradation (even more than 15%) with $K$ growing larger. Plausible as this phenomenon seems, it remains unclear whether it is inherent for local learning and how to alleviate this problem. In this section, we investigate the performance degradation issue of the greedy local training from an information-theoretic perspective, laying the basis for the proposed algorithm.

**Linear separability of intermediate features.** In the common case that greedy SL operates directly on the features output by internal layers, a natural intuition is to investigate how these locally learned features differ from their E2E learned counterparts in task-relevant behaviors. To this end, we fix the networks in Table 1, and train a linear classifier using the features from each layer. The test errors of these classifiers are presented in the left plot of Figure 2, where the horizontal axis denotes the indices of layers. The plot shows an intriguing trend: greedy SL contributes to dramatically more discriminative features with the first one (or few) local module, but is only able to slightly improve the performance with all the consequent modules. In contrast, the E2E learned network progressively boosts the linear separability of features throughout the whole network with even more significant effects in the later layers, surpassing greedy SL eventually. This raises an interesting question: *why does the full network achieve inferior performance in greedy SL compared to the E2E counterpart, even though the former is based on more discriminative earlier features?* This observation appears incompatible with prior works like deeply-supervised nets (Lee et al., 2015a).

**Information in features.** Since we use the same training configuration for both greedy SL and E2E learning, we conjecture that the answer to the above question might lie in the differences of features apart from merely separability. To test that, we look into the information captured by the intermediate features. In specific, given intermediate feature $\boldsymbol{h}$ corresponding to the input data $\boldsymbol{x}$ and the label $y$ (all of them are treated as random variables), we use the mutual information $I(\boldsymbol{h}, \boldsymbol{x})$ and $I(\boldsymbol{h}, y)$ to measure the amount of all retained information and task-relevant information in $\boldsymbol{h}$, respectively. As these metrics cannot be directly computed, we estimate the former by training a decoder with binary cross-entropy loss to reconstruct $\boldsymbol{x}$ from $\boldsymbol{h}$. For the latter, we train a CNN using $\boldsymbol{h}$ as inputs to correctly classify $\boldsymbol{x}$, and estimate $I(\boldsymbol{h}, y)$ with its performance. Details are deferred to Appendix G.

The estimates of $I(\boldsymbol{h}, \boldsymbol{x})$ and $I(\boldsymbol{h}, y)$ at different layers are shown in the middle and right plots of Figure 2. We note that in E2E learned networks, $I(\boldsymbol{h}, y)$ remains unchanged when the features pass through all the layers, while $I(\boldsymbol{h}, \boldsymbol{x})$ reduces gradually, revealing that the models progressively discard task-irrelevant information. However, greedily trained networks collapse both $I(\boldsymbol{h}, \boldsymbol{x})$ and $I(\boldsymbol{h}, y)$ in their first few modules. We attribute this to the short sighted optimization objective of earlier modules, which have relatively small capacity compared with full networks and are not capable of extracting and leveraging all the task-relevant information in $\boldsymbol{x}$, as the E2E learned networks

do. As a consequence, later modules, even though introducing additional parameters and increased capacity, lack necessary information about the target $y$ to construct more discriminative features.

**Information collapse hypothesis.** The above observations suggest that greedy SL induces local modules to collapse some of the task-relevant information that may be useless for short-term performance. However, the information is useful for the full model. In addition, we postulate that, although E2E training is incapable of extracting all task-relevant information at earlier layers as well, it alleviates this phenomenon by allowing a larger amount of task-irrelevant information to be kept, even though it may not be ideal for short-term performance. More empirical validation of our hypothesis is provided in Appendix A.

## 3 INFORMATION PROPAGATION (INFOPRO) LOSS

In this section, we propose an information propagation (InfoPro) loss to address the issue of information collapse in locally supervised training. The key idea is to enforce local modules to retain as much information about the input as possible, while progressively discard task-irrelevant parts. As it is difficult to estimate InfoPro loss in its exact form, we derive an easy-to-compute upper bound as the surrogate loss, and analyze its tightness.

### 3.1 LEARNING TO DISCARD USELESS INFORMATION

**Nuisance.** We first model the task-irrelevant information in the input data $x$ by introducing the concept of nuisance. A nuisance is defined as an arbitrary random variable that affects $x$ but provides no helpful information for the task of interest (Achille & Soatto, 2018). Take recognizing a car in the wild for example. The random variables determining the weather and the background are both nuisances. Formally, given a nuisance $r$, we have $I(r, x) > 0$ and $I(r, y) = 0$ , where $y$ is the label. Without loss of generality, we suppose that $y$, $r$, $x$ and $h$ form the Markov chain $(y, r) \rightarrow x \rightarrow h$, namely $p(h|x, y, r) = p(h|x)$. As a consequence, for the intermediate feature $h$ from any layer, we obviously have $I(r, x) \geq I(r, h)$. Nevertheless, we postulate that $\max_r I(r, h) > 0$. This assumption is mild since it does not hold only when $h$ strictly contains no task-irrelevant information.

**Information Propagation (InfoPro) Loss.** Thus far, we have been ready to introduce the proposed InfoPro loss. Instead of overly emphasizing on learning highly discriminative features at local modules, we also pay attention to preventing collapsing useful information in the feed-forward process. A simple solution to achieve this is maximizing the mutual information $I(h, x)$. Ideally, if there is no information loss, all useful information will be retained. However, it goes to another extreme case where the local modules do not learn any task-relevant feature, and is obviously dispensable. By contrast, in E2E training, intermediate layers progressively discard useless (task-irrelevant) information as well as shown above. Therefore, to model both effects simultaneously, we propose the following combined loss function:

$$\mathcal{L}_{\text{InfoPro}}(h) = \alpha[-I(h, x) + \beta I(r^*, h)], \ \alpha, \beta \geq 0, \quad s.t. \ r^* = \underset{r, \ I(r,x)>0, \ I(r,y)=0}{\operatorname{argmax}} I(r, h), \quad (1)$$

where the nuisance $r^*$ is formulated to capture as much task-irrelevant information in $h$ as possible, and the coefficient $\beta$ controls the amount of information that is propagated forward (first term) and task-irrelevant information that is discarded (second term). Notably, we assume that the final module is always trained using the normal E2E loss (e.g., softmax cross-entropy loss for classification) weighted by the constant 1, such that $\alpha$ is essential to balance the intermediate loss and the final one. In addition, $\mathcal{L}_{\text{InfoPro}}(h)$ is used to train the local module outputting $h$, whose inputs are not required to be $x$. The module may stack over another local module trained with the same form of $\mathcal{L}_{\text{InfoPro}}(h)$ but (possibly) different $\alpha$ and $\beta$.

Our method differs from existing works (Nøkland & Eidnes, 2019; Belilovsky et al., 2019; 2020) in that it is a non-greedy approach. The major effect of minimizing $\mathcal{L}_{\text{InfoPro}}(h)$ can be described as maximally discarding the task-irrelevant information under the goal of retaining as much information of the input as possible. Obtaining high short-term performance is not necessarily required. As we explicitly facilitate information propagation, we refer to $\mathcal{L}_{\text{InfoPro}}(h)$ as the InfoPro loss.

### 3.2 UPPER BOUND OF $\mathcal{L}_{\text{InfoPro}}$

The objective function in Eq.(1) is difficult to be directly optimized, since it is usually intractable to estimate $r^*$, which is equivalent to disentangling all task-irrelevant information from intermediate

features. Therefore, we derive an easy-to-compute upper bound of $\mathcal{L}_{\text{InfoPro}}$ as an surrogate loss. Our result is summarized in Proposition 1, with the proof in Appendix B.

**Proposition 1.** *Suppose that the Markov chain $(y, r) \to \boldsymbol{x} \to \boldsymbol{h}$ holds. Then an upper bound of $\mathcal{L}_{\text{InfoPro}}$ is given by*

$$\mathcal{L}_{\text{InfoPro}} \leq -\lambda_1 I(\boldsymbol{h}, \boldsymbol{x}) - \lambda_2 I(\boldsymbol{h}, y) \triangleq \bar{\mathcal{L}}_{\text{InfoPro}}, \tag{2}$$

*where $\lambda_1 = \alpha(1 - \beta)$, $\lambda_2 = \alpha\beta$.*

For simplicity, we integrate $\alpha$ and $\beta$ into two mutually independent hyper-parameters, $\lambda_1$ and $\lambda_2$. Although we do not explicitly restrict $\lambda_1 \geq 0$, we find in experiments that the performance of networks is significantly degraded with $\lambda_1 \to 0^+$ (see Figure 4), or say, $\beta \to 1^-$, where models tend to reach local minima by trivially minimizing $I(r^*, \boldsymbol{h})$ in Eq. (1). Thus, we assume $\lambda_1, \lambda_2 \geq 0$.

With Proposition 1, we can optimize the upper bound $\bar{\mathcal{L}}_{\text{InfoPro}}$ as an approximation, circumventing dealing with the intractable term $I(r^*, \boldsymbol{h})$ in $\mathcal{L}_{\text{InfoPro}}$. To ensure that the approximation is accurate, the gap between the two should be reasonably small. Below we present an analysis of the tightness of $\bar{\mathcal{L}}_{\text{InfoPro}}$ in Proposition 2, (proof given in Appendix C). We also empirically check it in Appendix H. Proposition 2 provides a useful tool to examine the discrepancy between $\mathcal{L}_{\text{InfoPro}}$ and its upper bound.

**Proposition 2.** *Given that $r^* = \operatorname{argmax}_{r, \, I(r, \boldsymbol{x}) > 0, \, I(r, y) = 0} I(r, \boldsymbol{h})$ and that $y$ is a deterministic function with respect to $\boldsymbol{x}$, the gap $\epsilon = \bar{\mathcal{L}}_{\text{InfoPro}} - \mathcal{L}_{\text{InfoPro}}$ is upper bounded by*

$$\epsilon \leq \lambda_2 \left[ I(\boldsymbol{x}, y) - I(\boldsymbol{h}, y) \right]. \tag{3}$$

### 3.3 Mutual Information Estimation

In the following, we describe the specific techniques we use to obtain the mutual information $I(\boldsymbol{h}, \boldsymbol{x})$ and $I(\boldsymbol{h}, y)$ in $\bar{\mathcal{L}}_{\text{InfoPro}}$. Both of them are estimated using small auxiliary networks. However, we note that the involved additional computational costs are minimal or even negligible (see Tables 3, 4).

**Estimating $I(\boldsymbol{h}, \boldsymbol{x})$.** Assume that $\mathcal{R}(\boldsymbol{x}|\boldsymbol{h})$ denotes the expected error for reconstructing $\boldsymbol{x}$ from $\boldsymbol{h}$. It has been widely known that $\mathcal{R}(\boldsymbol{x}|\boldsymbol{h})$ follows $I(\boldsymbol{h}, \boldsymbol{x}) = H(\boldsymbol{x}) - H(\boldsymbol{x}|\boldsymbol{h}) \geq H(\boldsymbol{x}) - \mathcal{R}(\boldsymbol{x}|\boldsymbol{h})$, where $H(\boldsymbol{x})$ denotes the marginal entropy of $\boldsymbol{x}$, as a constant (Vincent et al., 2008; Rifai et al., 2012; Kingma & Welling, 2013; Makhzani et al., 2015; Hjelm et al., 2019). Therefore, we estimate $I(\boldsymbol{h}, \boldsymbol{x})$ by training a decoder parameterized by $\boldsymbol{w}$ to obtain the minimal reconstruction loss, namely $I(\boldsymbol{h}, \boldsymbol{x}) \approx \max_{\boldsymbol{w}} [H(\boldsymbol{x}) - \mathcal{R}_{\boldsymbol{w}}(\boldsymbol{x}|\boldsymbol{h})]$. In practice, we use the binary cross-entropy loss for $\mathcal{R}_{\boldsymbol{w}}(\boldsymbol{x}|\boldsymbol{h})$.

**Estimating $I(\boldsymbol{h}, y)$.** We propose two ways to estimate $I(\boldsymbol{h}, y)$. Since $I(\boldsymbol{h}, y) = H(y) - H(y|\boldsymbol{h}) = H(y) - \mathbb{E}_{(\boldsymbol{h}, y)}[-\log p(y|\boldsymbol{h})]$, a straightforward approach is to train an auxiliary classifier $q_{\boldsymbol{\psi}}(y|\boldsymbol{h})$ with parameters $\boldsymbol{\psi}$ to approximate $p(y|\boldsymbol{h})$, such that we have $I(\boldsymbol{h}, y) \approx \max_{\boldsymbol{\psi}} \{H(y) - \mathbb{E}_{\boldsymbol{h}}[\sum_y -p(y|\boldsymbol{h})\log q_{\boldsymbol{\psi}}(y|\boldsymbol{h})]\}$. Note that this approximate equation will become an equation if and only if $q_{\boldsymbol{\psi}}(y|\boldsymbol{h}) \equiv p(y|\boldsymbol{h})$ (according to the Gibbs' inequality). Finally, we estimate the expectation on $\boldsymbol{h}$ using the samples $\{(\boldsymbol{x}_i, \boldsymbol{h}_i, y_i)\}_{i=1}^N$, namely $I(\boldsymbol{h}, y) \approx \max_{\boldsymbol{\psi}} \{H(y) - \frac{1}{N}[\sum_{i=1}^N -\log q_{\boldsymbol{\psi}}(y_i|\boldsymbol{h}_i)]\}$. Consequently, $q_{\boldsymbol{\psi}}(y|\boldsymbol{h})$ can be trained in a regular classification fashion with the cross-entropy loss.

In addition, motivated by recent advances in contrastive representation learning (Chen et al., 2020; Khosla et al., 2020; He et al., 2020), we formulate a contrastive style loss function $\mathcal{L}_{\text{contrast}}$, and prove in Appendix D that minimizing $\mathcal{L}_{\text{contrast}}$ is equivalent to maximizing a lower bound of $I(\boldsymbol{h}, y)$. Empirical results indicate that adopting $\mathcal{L}_{\text{contrast}}$ may lead to better performance if a large batch size is available. In specific, considering a mini-batch of intermediate features $\{\boldsymbol{h}_1, \ldots, \boldsymbol{h}_N\}$ corresponding to the labels $\{y_1, \ldots, y_N\}$, $\mathcal{L}_{\text{contrast}}$ is given by:

$$\mathcal{L}_{\text{contrast}} = \frac{1}{\sum_{i \neq j} \mathbb{1}_{y_i = y_j}} \sum_{i \neq j} \left[ \mathbb{1}_{y_i = y_j} \log \frac{\exp(\boldsymbol{z}_i^\top \boldsymbol{z}_j / \tau)}{\sum_{k=1}^N \mathbb{1}_{i \neq k} \exp(\boldsymbol{z}_i^\top \boldsymbol{z}_k / \tau)} \right], \quad \boldsymbol{z}_i = f_{\boldsymbol{\phi}}(\boldsymbol{h}_i). \tag{4}$$

Herein, $\mathbb{1}_{\mathbf{A}} \in \{0, 1\}$ returns 1 only when $\mathbf{A}$ is true, $\tau > 0$ is a pre-defined hyper-parameter, temperature, and $f_{\boldsymbol{\phi}}$ is a projection head parameterized by $\boldsymbol{\phi}$ that maps the feature $\boldsymbol{h}_i$ to a representation vector $\boldsymbol{z}_i$ (this design follows Chen et al. (2020); Khosla et al. (2020)).

**Implementation details.** We defer the details on the network architecture of $\boldsymbol{w}$, $\boldsymbol{\psi}$ and $\boldsymbol{\phi}$ to Appendix E. Briefly, on CIFAR, SVHN and STL-10, $\boldsymbol{w}$ is a two layer decoder with up-sampled inputs (if not otherwise noted), with $\boldsymbol{\psi}$ and $\boldsymbol{\phi}$ sharing the same architecture consisting of a single convolutional

layer followed by two fully-connected layers. On ImageNet and Cityscapes, we use relatively larger auxiliary nets, but they are very small compared with the primary network. Empirically, we find that these simple architectures are capable of achieving competitive performance consistently. Moreover, in implementation, we train $\boldsymbol{w}$, $\boldsymbol{\psi}$ and $\boldsymbol{\phi}$ collaboratively with the main network. Formally, let $\boldsymbol{\theta}$ denote the parameters of the local module to be trained, and then our optimization objective is

$$\underset{\boldsymbol{\theta},\boldsymbol{w},\boldsymbol{\psi}}{\text{minimize}} \ \lambda_1 \mathcal{R}_{\boldsymbol{w}}(\boldsymbol{x}|\boldsymbol{h}) + \lambda_2 \frac{1}{N}\sum_{i=1}^{N} -\log q_{\boldsymbol{\psi}}(y_i|\boldsymbol{h}_i) \ \text{ or } \ \underset{\boldsymbol{\theta},\boldsymbol{w},\boldsymbol{\phi}}{\text{minimize}} \ \lambda_1 \mathcal{R}_{\boldsymbol{w}}(\boldsymbol{x}|\boldsymbol{h}) + \lambda_2 \mathcal{L}_{\text{contrast}}, \quad (5)$$

which corresponds to using the cross-entropy and contrast loss to estimate $I(\boldsymbol{h}, y)$, respectively. Such an approximation is acceptable as we do not need to acquire the exact approximation of mutual information, and empirically it performs well in various experimental settings.

## 4 EXPERIMENTS

**Setups.** Our experiments are based on five widely used datasets (i.e., CIFAR-10 (Krizhevsky et al., 2009), SVHN (Netzer et al., 2011), STL-10 (Coates et al., 2011), ImageNet (Deng et al., 2009) and Cityscapes (Cordts et al., 2016)) and two popular network architectures (i.e., ResNet (He et al., 2016) and DenseNet (Huang et al., 2019)) with varying depth. We split each network into $K$ local modules with the same (or approximately the same) number of layers, where the first $K-1$ modules are trained using $\bar{\mathcal{L}}_{\text{InfoPro}}$, and the last module is trained using the standard E2E loss, as aforementioned. Due to spatial limitation, details on data pre-processing, training configurations and local module splitting are deferred to Appendix F. The hyper-parameters $\lambda_1$ and $\lambda_2$ are selected from $\{0, 0.1, 0.5, 1, 2, 5, 10, 20\}$. Notably, to avoid involving too many tunable hyper-parameters when $K$ is large (e.g., $K=16$), we assume that $\lambda_1$ and $\lambda_2$ change linearly from $1^{\text{st}}$ to $(K-1)^{\text{th}}$ local module, and thus we merely tune $\lambda_1$ and $\lambda_2$ for these two modules. We always use $\tau = 0.07$ in $\mathcal{L}_{\text{contrast}}$.

**Two training modes** are considered: (1) *simultaneous training*, where the back-propagation process of all local modules is sequentially triggered with every mini-batch of training data; and (2) *asynchronous training*, where local modules are isolatedly learned given cached outputs from completely trained earlier modules. Both modes enjoy high memory efficiency since only the activations within a single module require to be stored at a time. The second mode removes the dependence of local modules on their predecessors, enabling the fully decoupled training of network components. The experiments using asynchronous training are referred to as "Asy-InfoPro", while all other results are based on simultaneous training.

### 4.1 MAIN RESULTS

**Comparisons with other local learning methods.** We first compare the proposed InfoPro method with three recently proposed algorithms, decoupled greedy learning (Belilovsky et al., 2020) (DGL), Boost-ResNet (Huang et al., 2018a) and deep incremental boosting (Mosca & Magoulas, 2017) (DIB) in Figure 3. Our method yields the lowest test errors with all values of $K$. Notably, DGL can be viewed as a special case of InfoPro where $\lambda_1 = 0$. Hence, we use the same architecture of auxiliary networks as us in DGL for fair comparison. In addition, we present the estimates of mutual information between

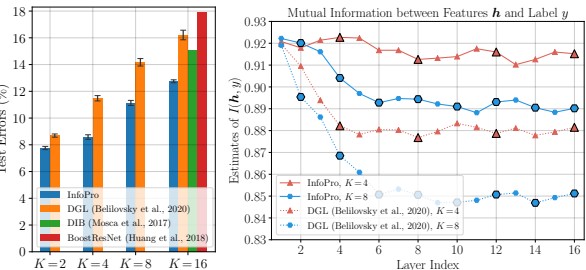

Figure 3: Comparisons of InfoPro and state-of-the-art local learning methods in terms of the test errors at the final layer (*left*) and the task-relevant information capture by intermediate features, $I(\boldsymbol{h}, y)$ (*right*). Results of ResNet-32 on CIFAR-10 are reported. We use the contrastive loss in $\bar{\mathcal{L}}_{\text{InfoPro}}$.

intermediate features and labels in the *right* plot of Figure 3. One can observe that DGL suffers from a severe collapse of task-relevant information at early modules, since it optimizes local modules greedily for merely short-term performance. By contrast, our method effectively alleviates this problem, retaining a larger amount of task-relevant information within intermediate features.

**Results on various image classification benchmarks** are presented in Table 2. We also report the result of DGL (Belilovsky et al., 2020) in our implementation. It can be observed that InfoPro outperforms greedy SL by large margins consistently across different networks, especially when $K$ is

Table 2: Performance of different networks with varying numbers of local modules. The averaged test errors and standard deviations of 5 independent trials are reported. InfoPro (Softmax/Contrast) refers to two approaches to estimating $I(\boldsymbol{h}, y)$. The results of Asy-InfoPro is obtain by asynchronous training, while others are based on simultaneous training. Greedy SL$^+$ adopts deeper networks to have the same computational costs as InfoPro.

| Dataset | Network | Method | $K = 2$ | $K = 4$ | $K = 8$ | $K = 16$ |
|---|---|---|---|---|---|---|
| CIFAR-10 | ResNet-32 (E2E: 7.37 ± 0.10%) | Greedy SL | 10.30 ± 0.20% | 16.07 ± 0.46% | 21.19 ± 0.52% | 24.59 ± 0.83% |
| | | DGL (Belilovsky et al., 2020) | 8.69 ± 0.12% | 11.48 ± 0.20% | 14.17 ± 0.28% | 16.21 ± 0.36% |
| | | InfoPro (Softmax) | 8.13 ± 0.23% | 8.64 ± 0.25% | 11.40 ± 0.18% | 14.23 ± 0.42% |
| | | InfoPro (Contrast) | **7.76 ± 0.12%** | **8.58 ± 0.17%** | **11.13 ± 0.19%** | **12.75 ± 0.11%** |
| | ResNet-110 (E2E: 6.50 ± 0.34%) | Greedy SL | 8.21 ± 0.24% | 13.16 ± 0.28% | 15.61 ± 0.57% | 18.92 ± 1.27% |
| | | Greedy SL$^+$ | 8.00 ± 0.11% | 12.47 ± 0.17% | 14.58 ± 0.36% | 17.35 ± 0.31% |
| | | DGL (Belilovsky et al., 2020) | 7.70 ± 0.28% | 10.50 ± 0.11% | 12.46 ± 0.37% | 13.80 ± 0.15% |
| | | InfoPro (Softmax) | 7.01 ± 0.34% | 7.96 ± 0.06% | 9.40 ± 0.27% | 10.78 ± 0.28% |
| | | Asy-InfoPro (Contrast) | 7.34 ± 0.11% | 8.39 ± 0.15% | – | – |
| | | InfoPro (Contrast) | **6.42 ± 0.08%** | **7.30 ± 0.14%** | **8.93 ± 0.40%** | **9.90 ± 0.19%** |
| | DenseNet-BC-100-12 (E2E: 4.61 ± 0.08%) | Greedy SL | 5.10 ± 0.05% | 6.07 ± 0.21% | 8.21 ± 0.31% | 10.41 ± 0.42% |
| | | DGL (Belilovsky et al., 2020) | 4.86 ± 0.15% | 5.71 ± 0.04% | 6.82 ± 0.21% | 7.67 ± 0.16% |
| | | InfoPro (Softmax) | 4.79 ± 0.07% | 5.69 ± 0.21% | 6.44 ± 0.11% | 7.47 ± 0.21% |
| | | InfoPro (Contrast) | **4.74 ± 0.04%** | **5.24 ± 0.25%** | **5.86 ± 0.18%** | **6.92 ± 0.16%** |
| SVHN | ResNet-110 (E2E: 3.07 ± 0.23%) | Greedy SL | 3.71 ± 0.16% | 5.39 ± 0.22% | 5.75 ± 0.10% | 6.37 ± 0.42% |
| | | DGL (Belilovsky et al., 2020) | 3.61 ± 0.16% | 4.97 ± 0.19% | 5.35 ± 0.13% | 5.55 ± 0.34% |
| | | InfoPro (Softmax) | 3.41 ± 0.08% | 3.72 ± 0.03% | 4.67 ± 0.07% | 5.14 ± 0.08% |
| | | InfoPro (Contrast) | **3.15 ± 0.03%** | **3.28 ± 0.11%** | **3.62 ± 0.11%** | **3.91 ± 0.16%** |
| STL-10 | ResNet-110 (E2E: 22.27 ± 1.61%) | Greedy SL | 25.56 ± 1.37% | 27.97 ± 0.75% | 29.07 ± 0.76% | 30.38 ± 0.39% |
| | | DGL (Belilovsky et al., 2020) | 24.96 ± 1.18% | 26.77 ± 0.64% | 27.33 ± 0.24% | 27.73 ± 0.58% |
| | | InfoPro (Softmax) | 21.02 ± 0.51% | **21.28 ± 0.27%** | **23.60 ± 0.49%** | **26.05 ± 0.71%** |
| | | InfoPro (Contrast) | **20.99 ± 0.64%** | 22.73 ± 0.40% | 25.15 ± 0.52% | 26.27 ± 0.48% |

Table 3: Trade-off between GPU memory footprint during training and test errors. Results of training ResNet-110 on a single Nvidia Titan Xp GPU are reported. 'GC' refers to gradient checkpointing (Chen et al., 2016).

| Methods | CIFAR-10 (batch size = 1024) | | | STL-10 (batch size = 128) | | |
|---|---|---|---|---|---|---|
| | Test Error | Memory Cost | Computational Overhead (Theoretical / Wall Time) | Test Error | Memory Cost | Computational Overhead (Theoretical / Wall Time) |
| E2E Training | 6.50 ± 0.34% | 9.40 GB | – | 22.27 ± 1.61% | 10.77 GB | – |
| GC (Chen et al., 2016) | 6.50 ± 0.34% | 3.91 GB (↓58.4%) | 32.8% / 27.5% | 22.27 ± 1.61% | 4.50 GB (↓58.2%) | 32.8% / 27.0% |
| InfoPro$^*$, $K = 2$ | **6.41 ± 0.13%** | 5.38 GB (↓42.8%) | **1.3% / 1.1%** | **20.95 ± 0.57%** | 6.15 GB (↓42.9%) | **1.3% / 1.7%** |
| InfoPro$^*$, $K = 3$ | 6.74 ± 0.12% | 4.22 GB (↓55.1%) | 3.3% / 7.5% | 21.00 ± 0.52% | 4.96 GB (↓53.9%) | 3.3% / 7.0% |
| InfoPro$^*$, $K = 4$ | 6.93 ± 0.20% | **3.52 GB (↓62.6%)** | 5.9% / 13.4% | 21.22 ± 0.72% | **4.08 GB (↓62.1%)** | 5.9% / 11.4% |

Table 4: Single crop error rates (%) on the validation set of ImageNet. We use 8 Tesla V100 GPUs for training.

| Models | Methods | Batch Size | Top-1 Error | Top-5 error | Memory Cost (per GPU) | Computational Overhead (Theoretical / Wall Time) |
|---|---|---|---|---|---|---|
| ResNet-101 | E2E Training | 1024 | 22.03% | 5.93% | 19.71 GB | – |
| | InfoPro$^*$, $K = 2$ | 1024 | **21.85%** | **5.89%** | **12.06 GB (↓38.8%)** | 5.7% / 11.7% |
| ResNet-152 | E2E Training | 1024 | 21.60% | 5.92% | 26.29 GB | – |
| | InfoPro$^*$, $K = 2$ | 1024 | **21.45%** | **5.84%** | **15.53 GB (↓40.9%)** | 3.9% / 8.7% |
| ResNeXt-101, 32×8d | E2E Training | 512 | 20.64% | 5.40% | 19.22 GB | – |
| | InfoPro$^*$, $K = 2$ | 512 | **20.35%** | **5.28%** | **11.55 GB (↓39.9%)** | 2.7% / 5.6% |

large. For example, on CIFAR-10, ResNet-32 + InfoPro achieves a test error of $12.75\%$ with $K = 16$, surpassing greedy SL by $11.84\%$. For ResNet-110, we note that our method performs on par with E2E training with $K = 2$, while degrading the performance by up to $0.8\%$ with $K = 4$. Moreover, InfoPro is shown to compare favorably against DGL under most settings.

In addition, given that our method introduces auxiliary networks, we enlarge network depth for greedy SL to match the computational cost of InfoPro, named as greedy SL$^+$. However, this only slightly ameliorates the performance since the problem of information collapse still exists. Another interesting phenomenon is that InfoPro (Contrast) outperforms InfoPro (Softmax) on CIFAR-10 and SVHN, yet fails to do so on STL-10. We attribute this to the larger batch size we use on the former two datasets and the proper value of the temperature $\tau$. A detailed analysis is given in Appendix H.

**Asynchronous and parallel training.** The results of asynchronous training are presented in Table 2 as "Asy-InfoPro", and it appears to slightly hurt the performance. Asy-InfoPro differentiates itself from InfoPro on that it adopts the cached outputs from completely trained earlier modules as the inputs of later modules. Therefore, the degradation of performance might be ascribed to lacking regularizing effects from the noisy outputs of earlier modules during training (Löwe et al., 2019). However, Asy-InfoPro is still considerably better than both greedy SL and DGL, approaching E2E

Table 5: Results of semantic segmentation on Cityscapes. 2 Nvidia GeForce RTX 3090 GPUs are used for training. 'SS' refers to the single-scale inference. 'MS' and 'Flip' denote employing the average prediction of multi-scale ([0.5, 1.75]) and left-right flipped inputs during inference. We also present the results reported by the original paper in the "original" row. DGL refers to decoupled greedy learning (Belilovsky et al., 2020).

| Model | Training Algorithms | Training Iterations | Batch Size | Crop Size | mIoU | | | Memory Cost (per GPU) | Computational Overhead (Theoretical / Wall Time) |
|---|---|---|---|---|---|---|---|---|---|
| | | | | | SS | MS | MS+Flip | | |
| DeepLab-V3 -R101 (w/ syncBN) | E2E (original) | 40k | 8 | 769×769 | 77.82% | 79.06% | 79.30% | – | |
| | E2E (ours) | 40k | 8 | 512×1024 | 79.12% | 79.81% | 80.02% | 19.43GB | – |
| | DGL | 40k | 8 | 512×1024 | 78.15% | 79.40% | 79.56% | – | – |
| | InfoPro*, $K=2$ | 40k | 8 | 512×1024 | **79.37%** | **80.53%** | **80.54%** | **12.01GB** ($\downarrow 38.2\%$) | 6.4% / 2.2% |
| | E2E (ours) | 60k | 8 | 512×1024 | 79.32% | 79.95% | 80.07% | 19.43GB | – |
| | InfoPro*, $K=2$ | **40k** | 12 | 512×1024 | 79.99% | 81.09% | 81.20% | **16.62GB** ($\downarrow 14.5\%$) | 6.4% / $\downarrow 2.1\%$ |
| | InfoPro*, $K=2$ | **40k** | 8 | **640×1280** | **80.25%** | **81.33%** | **81.42%** | 17.00GB ($\downarrow 12.5\%$) | 10.3% / $\downarrow 2.9\%$ |

training. Besides, we note that asynchronous training can be easily extended to training different local modules in parallel by dynamically caching the outputs of earlier modules. To this end, we preliminarily test training two local modules parallelly on 2 GPUs when $K = 2$, using the same experimental protocols as Huo et al. (2018b) (train ResNet-110 with a batch size of 128 on CIFAR-10) and their public code. Our method gives a 1.5× speedup over the standard parallel paradigm of E2E training (the DataParallel toolkit in Pytorch). Note that parallel training has *the same* performance as simultaneous training (i.e., "InfoPro" in Table 2) since their training processes are identical except for the parallelism.

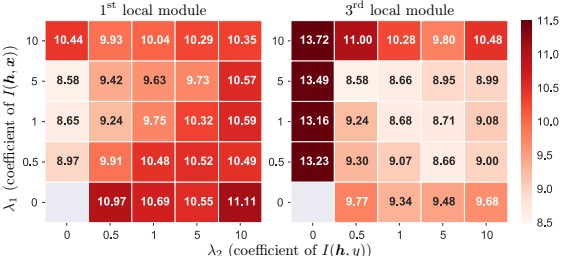

Figure 4: Sensitivity tests. The CIFAR-10 test errors of ResNet-32 trained using InfoPro ($K=4$) are reported. We vary $\lambda_1$ and $\lambda_2$ for 1$^{\text{st}}$ and 3$^{\text{rd}}$ local modules respectively, with all other modules unchanged. We do not consider $\lambda_1 = \lambda_2 = 0$, where we obviously have $\bar{\mathcal{L}}_{\text{InfoPro}} = \mathcal{L}_{\text{InfoPro}} \equiv 0$.

**Reducing GPUs memory requirements.** Here we split the network into local modules to ensure each module consumes a similar amount of GPU memory during training. Note that this is different from splitting the model into modules with the same number of layers. We denote the results in this setting by InfoPro*, and the trade-off between GPU memory consumption and test errors is presented in Table 3, where we report the minimally required GPU memory to run the training algorithm. The contrastive and softmax loss are used in InfoPro* on CIFAR-10 and STL-10, respectively. One can observe that our method significantly improves the memory efficiency of CNNs. For instance, on STL-10, InfoPro* ($K = 4$) outperforms the E2E baseline by $1.05\%$ with 37.9% of the GPUs memory requirements. The computational overhead is presented in both the theoretical results and the practical wall time. Due to implementation issues, we find that the latter is slightly larger than the former for InfoPro*. Compared to the gradient checkpointing technique (Chen et al., 2016), our method achieves competitive performance with significantly reduced computational and time cost.

**Results on ImageNet** are reported in Table 4. The softmax loss is used in InfoPro* since the batch size is relatively small. The proposed method reduces the memory cost by 40%, and achieves slightly better performance. Notably, our method enables training these large networks using 16 GB GPUs.

**Results of semantic segmentation on Cityscapes** are presented in Table 5. We report the mean Intersection over Union (mIoU) of all classes on the validation set. The softmax loss is used in InfoPro*. The details of $\psi$ and $w$ are presented in Appendix E. Our method boosts the performance of the DeepLab-V3 (Chen et al., 2017) network and allows training the model with 50% larger batch sizes ($2 \rightarrow 3$ per GPU) under the same memory constraints. This contributes to more accurate statistics for batch normalization, which is a practical requirement for tasks with high resolution inputs. In addition, InfoPro* enables using larger crop sizes (512×1024 $\rightarrow$ 640×1280) during training without enlarging GPUs memory footprint, which significantly improves the mIoU. Note that this does not increase the training or inference cost.

## 4.2 Hyper-parameter Sensitivity and ablation study

**The coefficient $\lambda_1$ and $\lambda_2$.** To study how $\lambda_1$ and $\lambda_2$ affect the performance, we change them for the 1$^{\text{st}}$ and 3$^{\text{rd}}$ local modules of a ResNet-32 trained using InfoPro (Contrast), $K = 4$, with the results shown in Figure 4. We find that the earlier module benefits from small $\lambda_2$ to propagate more information forward, while larger $\lambda_2$ helps the later module to boost the final accuracy. This is

compatible with previous works showing that removing earlier layers in ResNets has a minimal impact on performance (Veit et al., 2016).

**Ablation study**. For ablation, we test directly removing the decoder $w$ or replacing the contrastive head $\phi$ by the linear classifier used in greedy SL, as shown in Table 6.

Table 6: Ablation studies. Test errors of ResNet-32 on CIFAR-10 are reported.

| $w$ | $\phi$ | $K = 2$ | $K = 8$ |
|---|---|---|---|
| | | $10.30 \pm 0.20\%$ | $21.19 \pm 0.52\%$ |
| | ✓ | $8.90 \pm 0.17\%$ | $15.82 \pm 0.34\%$ |
| ✓ | | $8.49 \pm 0.16\%$ | $14.13 \pm 0.22\%$ |
| ✓ | ✓ | $\mathbf{7.76 \pm 0.12\%}$ | $\mathbf{11.13 \pm 0.19\%}$ |

## 5 RELATED WORK

**Greedy training of deep networks** is first proposed to learn unsupervised deep generative models, or to obtain an appropriate initialization for E2E supervised training (Hinton et al., 2006; Bengio et al., 2007). However, later works reveal that this initialization is indeed dispensable once proper networks architectures are adopted, e.g., introducing batch normalization layers (Ioffe & Szegedy, 2015), skip connections (He et al., 2016) or dense connections (Huang et al., 2019). Some other works (Kulkarni & Karande, 2017; Malach & Shalev-Shwartz, 2018; Marquez et al., 2018; Huang et al., 2018a) attempt to learn deep models in a layer-wise fashion. For example, BoostResNet (Huang et al., 2018a) trains different residual blocks in a ResNet (He et al., 2016) sequentially with a boosting algorithm. Deep Cascade Learning (Marquez et al., 2018) extends the cascade correlation algorithm (Fahlman & Lebiere, 1990) to deep learning, aiming at improving the training efficiency. However, these approaches mainly focus on theoretical analysis and are usually validated with limited experimental results on small datasets. More recently, several works have pointed out the inefficiencies of back-propagation and revisited this problem (Nøkland & Eidnes, 2019; Belilovsky et al., 2019; 2020). These works adopt a similar local learning setting to us, while they mostly optimize local modules with a greedy short-term objective, and hence suffer from the information collapse issue we discuss in this paper. In contrast, our method trains local modules by minimizing the non-greedy InfoPro loss.

**Alternatives of back-propagation** have been widely studied in recent years. Some biologically-motivated algorithms including target propagation (Lee et al., 2015b; Bartunov et al., 2018) and feedback alignment (Lillicrap et al., 2014; Nøkland, 2016) avoid back-propagation by directly propagating backward optimal activations or error signals with auxiliary networks. Decoupled Neural Interfaces (DNI) (Jaderberg et al., 2017) learn auxiliary networks to produce synthetic gradients. In addition, optimization methods like Alternating Direction Method of Multipliers (ADMM) split the end-to-end optimization into sub-problems using auxiliary variables (Taylor et al., 2016; Choromanska et al., 2018). Decoupled Parallel Back-propagation (Huo et al., 2018b) and Features Replay (Huo et al., 2018a) update parameters with previous gradients instead of current ones, and show its convergence theoretically, enabling training network modules in parallel. Nevertheless, these methods are fundamentally different from us as they train local modules by explicitly or implicitly optimizing the global objective, while we merely consider optimizing *local* objectives.

**Information-theoretic analysis in deep learning** has received increasingly more attention in the past few years. Shwartz-Ziv & Tishby (2017) and Saxe et al. (2019) study the information bottleneck (IB) principle (Tishby et al., 2000) to explain the training dynamics of deep networks. Achille & Soatto (2018) decompose the cross-entropy loss and propose a novel IB for weights. There are also efforts towards fulfilling efficient training with IB (Alemi et al., 2016). In the context of unsupervised learning, a number of methods have been proposed based on mutual information maximization (Oord et al., 2018; Tian et al., 2020; Hjelm et al., 2019). SimCLR (Chen et al., 2020) and MoCo (He et al., 2020) propose to maximize the mutual information of different views from the same input with the contrastive loss. This paper analyzes the drawbacks of greedy local supervision and propose the InfoPro loss from the information-theoretic perspective as well. In addition, our method can also be implemented as the combination of a contrastive term and a reconstruction loss.

## 6 CONCLUSION

This work studied locally supervised deep learning from the information-theoretic perspective. We demonstrated that training local modules greedily results in collapsing task-relevant information at earlier layers, degrading the final performance. To address this issue, we proposed an information propagation (InfoPro) loss that encourages local modules to preserve more information about the input, while progressively discard task-irrelevant information. Extensive experiments validated that InfoPro significantly reduced GPUs memory footprint during training without sacrificing accuracy. It also enabled model parallelization in an asynchronous fashion. InfoPro may open new avenues for developing more efficient and biologically plausible deep learning algorithms.

ACKNOWLEDGMENTS

This work is supported in part by the National Science and Technology Major Project of the Ministry of Science and Technology of China under Grants 2018AAA0100701, the National Natural Science Foundation of China under Grants 61906106 and 62022048, the Institute for Guo Qiang of Tsinghua University and Beijing Academy of Artificial Intelligence.

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

APPENDIX

## A    A TOY EXAMPLE

To further validate the proposed information collapse hypothesis, we visualize the "information flow" within deep networks using a toy example. First, we establish a MNIST-STL10 dataset via placing MNIST digits on a certain position (randomly selected from 64 candidates) of a background image from STL-10. Then, three specific tasks can be defined on MNIST-STL10, namely classifying digits, backgrounds and positions of numbers. We refer to the labels of them as $y_1$, $y_2$ and $y_3$, respectively, as illustrated by Figure 5.

| Input $\boldsymbol{x}$ | Labels |
|---|---|
| | $y_1$: Background |
| | $y_2$: Digit |
| | $y_3$: Position of Digit |

Figure 5: Illustration of the MNIST-STL10 dataset.

We train ResNet-32 networks for the three tasks with greedy SL ($K=4$) and end-to-end training ($K=1$). The estimates of mutual information $I(\boldsymbol{h}, y_1)$, $I(\boldsymbol{h}, y_2)$ and $I(\boldsymbol{h}, y_3)$ are shown in Figure 6, with the same estimating approach as Figure 2 (details in Appendix G). Note that when one label (take $y_1$ for example) is adopted for training, the information related to other labels ($I(\boldsymbol{h}, y_2)$ and $I(\boldsymbol{h}, y_3)$) is task-irrelevant. From the plots, one can clearly observe that end-to-end training retains all task-relevant information throughout the feed-forward process, while greedy SL usually yields less informative intermediate representations in terms of the task of interest. This phenomenon confirms the proposed information collapse hypothesis empirically. In addition, we postulate that the end-to-end learned early layers prevent collapsing task-relevant information by being allowed to keep larger amount of task-irrelevant information, which, however, may lead to the inferior classification performance of intermediate features, and thus cannot be achieved by greedy SL.

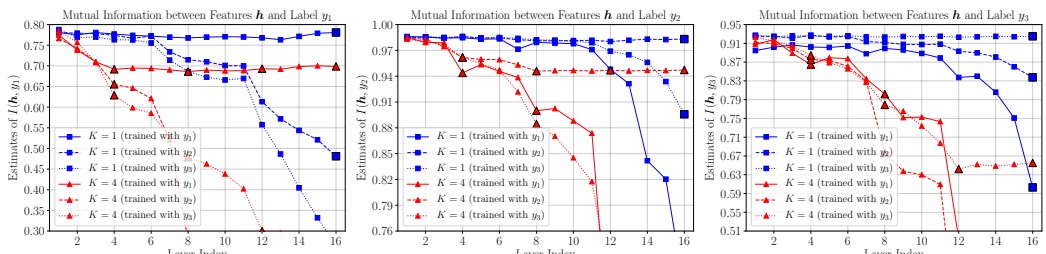

Figure 6: The estimates of mutual information between the intermediate features $\boldsymbol{h}$ and the three labels of MNIST-STL10 (see: Figure 5), i.e. $y_1$ (*left*, background), $y_2$ (*middle*, digit) and $y_3$ (*right*, position of digit). Models are trained using greedy SL ($K = 1, 4$) supervised by one of the three labels, and the results are shown with respect to layer indices. "$K = 1$" refers to end-to-end training.

## B    PROOF OF PROPOSITION 1

**Proposition 1.** *Suppose that the Markov chain* $(y, r) \to \boldsymbol{x} \to \boldsymbol{h}$ *holds. Then an upper bound of* $\mathcal{L}_{\text{InfoPro}}$ *is given by*

$$\mathcal{L}_{\text{InfoPro}} \leq -\lambda_1 I(\boldsymbol{h}, \boldsymbol{x}) - \lambda_2 I(\boldsymbol{h}, y) \triangleq \bar{\mathcal{L}}_{\text{InfoPro}}, \tag{6}$$

*where* $\lambda_1 = \alpha(1 - \beta)$, $\lambda_2 = \alpha\beta$.

**Proof.** Note that, $\mathcal{L}_{\text{InfoPro}}$ is given by

$$\mathcal{L}_{\text{InfoPro}}(\boldsymbol{h}) = \alpha[-I(\boldsymbol{h}, \boldsymbol{x}) + \beta I(r^*, \boldsymbol{h})], \ \alpha, \beta \geq 0, \quad s.t. \ r^* = \underset{r, \ I(r, \boldsymbol{x}) > 0, \ I(r, y) = 0}{\operatorname{argmax}} I(r, \boldsymbol{h}). \tag{7}$$

Due to the Markov chain $(y, r) \to \boldsymbol{x} \to \boldsymbol{h}$, we have $I(\boldsymbol{h}, (y, r^*)) \leq I(\boldsymbol{h}, \boldsymbol{x})$. Given that

$$\begin{aligned}
I(\boldsymbol{h}, (y, r^*)) &= H(\boldsymbol{h}) - H(\boldsymbol{h}|y, r^*) \\
&= [H(\boldsymbol{h}) - H(\boldsymbol{h}|r^*)] + [H(\boldsymbol{h}|r^*) - H(\boldsymbol{h}|y, r^*)] \\
&= I(\boldsymbol{h}, r^*) + I(\boldsymbol{h}, y|r^*),
\end{aligned} \tag{8}$$

we have

$$I(\boldsymbol{h}, r^*) \leq I(\boldsymbol{h}, \boldsymbol{x}) - I(\boldsymbol{h}, y|r^*). \tag{9}$$

By the definition of nuisance, we note that $r^*$ and $y$ are mutually independent, and thus we obtain

$$
\begin{aligned}
I(\boldsymbol{h}, y | r^*) &= H(y | r^*) - H(y | \boldsymbol{h}, r^*) \\
&= H(y) - H(y | \boldsymbol{h}, r^*) \\
&\geq H(y) - H(y | \boldsymbol{h}) = I(\boldsymbol{h}, y).
\end{aligned} \tag{10}
$$

Combining Eqs. (9) and (10), we have

$$
I(\boldsymbol{h}, r^*) \leq I(\boldsymbol{h}, \boldsymbol{x}) - I(\boldsymbol{h}, y). \tag{11}
$$

Finally, Proposition 1 is proved by combining Eq. (7) and Inequality (11). □

## C  PROOF OF PROPOSITION 2

We first introduce a Lemma proven by Achille & Soatto (2018).

**Lemma 1.** *Given a joint distribution $p(\boldsymbol{x}, y)$, where $y$ is a discrete random variable, we can always find a random variable $r$ independent of $y$ such that $\boldsymbol{x} = f(y, r)$, for some deterministic function $f$.*

**Proposition 2.** *Given that $r^* = \operatorname{argmax}_{r, I(r, \boldsymbol{x}) > 0, I(r, y) = 0} I(r, \boldsymbol{h})$ and that $y$ is a deterministic function with respect to $\boldsymbol{x}$, the gap $\epsilon = \bar{\mathcal{L}}_{\text{InfoPro}} - \mathcal{L}_{\text{InfoPro}}$ is upper bounded by*

$$
\epsilon \leq \lambda_2 \left[ I(\boldsymbol{x}, y) - I(\boldsymbol{h}, y) \right]. \tag{12}
$$

*Proof.* Let $\tilde{r}$ be the random variable in Lemma 1, and then, since we can find a deterministic function that maps $\boldsymbol{x}$ to $y$, we have

$$
I(\boldsymbol{h}, \boldsymbol{x}) = I(\boldsymbol{h}, (y, \tilde{r})) = I(\boldsymbol{h}, \tilde{r}) + I(\boldsymbol{h}, y | \tilde{r}). \tag{13}
$$

Assume $\zeta = [I(\boldsymbol{h}, \boldsymbol{x}) - I(\boldsymbol{h}, y)] - I(\boldsymbol{h}, r^*)$, in terms that $r^* = \operatorname{argmax}_{r, I(r, \boldsymbol{x}) > 0, I(r, y) = 0} I(r, \boldsymbol{h})$, we obtain

$$
\begin{aligned}
\zeta &\leq [I(\boldsymbol{h}, \boldsymbol{x}) - I(\boldsymbol{h}, y)] - I(\boldsymbol{h}, \tilde{r}) \\
&= I(\boldsymbol{h}, \tilde{r}) + I(\boldsymbol{h}, y | \tilde{r}) - I(\boldsymbol{h}, y) - I(\boldsymbol{h}, \tilde{r}) \\
&= I(\boldsymbol{h}, y | \tilde{r}) - I(\boldsymbol{h}, y) \\
&= I(\boldsymbol{h}, y | \tilde{r}) - I(\boldsymbol{x}, y) + I(\boldsymbol{x}, y) - I(\boldsymbol{h}, y) \\
&= H(y | \tilde{r}) - H(y | \tilde{r}, \boldsymbol{h}) - H(y) + H(y | \boldsymbol{x}) + I(\boldsymbol{x}, y) - I(\boldsymbol{h}, y).
\end{aligned} \tag{14}
$$

Since $y$ and $\tilde{r}$ are mutually independent, namely $H(y | \tilde{r}) = H(y)$, we have

$$
\begin{aligned}
\zeta &\leq H(y) - H(y | \tilde{r}, \boldsymbol{h}) - H(y) + H(y | \boldsymbol{x}) + I(\boldsymbol{x}, y) - I(\boldsymbol{h}, y) \\
&= H(y | \boldsymbol{x}) - H(y | \tilde{r}, \boldsymbol{h}) + I(\boldsymbol{x}, y) - I(\boldsymbol{h}, y) \\
&\leq H(y | \boldsymbol{x}) + I(\boldsymbol{x}, y) - I(\boldsymbol{h}, y).
\end{aligned} \tag{15}
$$

When considering $y$ as a deterministic function with regards to $\boldsymbol{x}$, we obtain $H(y | \boldsymbol{x}) = 0$, and therefore

$$
\zeta \leq I(\boldsymbol{x}, y) - I(\boldsymbol{h}, y). \tag{16}
$$

Given that $\epsilon = \bar{\mathcal{L}}_{\text{InfoPro}} - \mathcal{L}_{\text{InfoPro}} = \lambda_2 \zeta$, we have

$$
\epsilon \leq \lambda_2 \left[ I(\boldsymbol{x}, y) - I(\boldsymbol{h}, y) \right], \tag{17}
$$

for which we have proven Proposition 2. □

## D  WHY MINIMIZING THE CONTRASTIVE LOSS MAXIMIZES THE LOWER BOUND OF TASK-RELEVANT INFORMATION?

In this section, we show that minimizing the proposed contrastive loss, namely

$$
\mathcal{L}_{\text{contrast}} = \frac{1}{\sum_{i \neq j} \mathbb{1}_{y_i = y_j}} \sum_{i \neq j} \left[ \mathbb{1}_{y_i = y_j} \log \frac{\exp(\boldsymbol{z}_i^\top \boldsymbol{z}_j / \tau)}{\sum_{k=1}^N \mathbb{1}_{i \neq k} \exp(\boldsymbol{z}_i^\top \boldsymbol{z}_k / \tau)} \right], \quad \boldsymbol{z}_i = f_\phi(\boldsymbol{h}_i), \tag{18}
$$

actually maximizes an lower bound of task-relevant information $I(\boldsymbol{h}, y)$. We start by considering a simplified but equivalent situation. Suppose that we have a query sample $\boldsymbol{z}^+$, together with a set

$X = \{z_1, \ldots, z_N\}$ consisting of $N$ samples with one positive sample $z^{\mathrm{p}}$ from the same class as $z^+$ definitely, and other negative samples are randomly sampled, namely $X = \{z^{\mathrm{p}}\} \cup X_{\mathrm{neg}}$. Then the expectation of $\mathcal{L}_{\mathrm{contrast}}$ can be written as

$$\mathbb{E}[\mathcal{L}_{\mathrm{contrast}}] = \mathbb{E}_{z^+, X}\left[-\log \frac{\exp(z^{+\top}z^{\mathrm{p}}/\tau)}{\sum_{i=1}^{N} \exp(z^{+\top}z_i/\tau)}\right]. \tag{19}$$

Eq. (19) can be viewed as a categorical cross-entropy loss of recognizing the positive sample $z^{\mathrm{p}}$ correctly. Hence, we define the optimal probability of this classification problem as $P^{\mathrm{pos}}(z_i|X)$, which denotes the true probability of $z_i$ being the positive sample. Assuming that the label of $z^{\mathrm{p}}$ is $y$, the positive and negative samples can be viewed as being sampled from the true distributions $p(z|y)$ and $p(z)$, respectively. As a consequence, $P^{\mathrm{pos}}(z_i|X)$ can be derived as

$$P^{\mathrm{pos}}(z_i|X) = \frac{p(z_i|y)\prod_{l\neq i}p(z_l)}{\sum_{j=1}^{N}p(z_j|y)\prod_{l\neq j}p(z_l)} = \frac{\frac{p(z_i|y)}{p(z_i)}}{\sum_{j=1}^{N}\frac{p(z_j|y)}{p(z_j)}}, \tag{20}$$

which indicates that an optimal value for $\exp(z^{+\top}z^{\mathrm{p}}/\tau)$ is $\frac{p(z^{\mathrm{p}}|y)}{p(z^{\mathrm{p}})}$. Therefore, by assuming that $z^+$ is uniformly sampled from all classes, we have

$$\mathbb{E}[\mathcal{L}_{\mathrm{contrast}}] \geq \mathbb{E}[\mathcal{L}_{\mathrm{contrast}}^{\mathrm{optimal}}] = \mathbb{E}_{y,X}\left[-\log \frac{\frac{p(z^{\mathrm{p}}|y)}{p(z^{\mathrm{p}})}}{\sum_{j=1}^{N}\frac{p(z_j|y)}{p(z_j)}}\right] \tag{21}$$

$$= \mathbb{E}_{y,X}\left[-\log \frac{\frac{p(z^{\mathrm{p}}|y)}{p(z^{\mathrm{p}})}}{\frac{p(z^{\mathrm{p}}|y)}{p(z^{\mathrm{p}})} + \sum_{z_j \in X_{\mathrm{neg}}}\frac{p(z_j|y)}{p(z_j)}}\right] \tag{22}$$

$$= \mathbb{E}_{y,X}\left\{\log\left[1 + \frac{p(z^{\mathrm{p}})}{p(z^{\mathrm{p}}|y)}\sum_{z_j \in X_{\mathrm{neg}}}\frac{p(z_j|y)}{p(z_j)}\right]\right\} \tag{23}$$

$$\approx \mathbb{E}_{y,X}\left\{\log\left[1 + \frac{p(z^{\mathrm{p}})}{p(z^{\mathrm{p}}|y)}(N-1)\mathbb{E}_{z_j \sim p(z_j)}\frac{p(z_j|y)}{p(z_j)}\right]\right\} \tag{24}$$

$$= \mathbb{E}_{y,z^{\mathrm{p}}}\left\{\log\left[1 + \frac{p(z^{\mathrm{p}})}{p(z^{\mathrm{p}}|y)}(N-1)\right]\right\} \tag{25}$$

$$\geq \mathbb{E}_{y,z^{\mathrm{p}}}\left\{\log\left[\frac{p(z^{\mathrm{p}})}{p(z^{\mathrm{p}}|y)}(N-1)\right]\right\} \tag{26}$$

$$= -I(z^{\mathrm{p}}, y) + \log(N-1) \geq -I(h, y) + \log(N-1). \tag{27}$$

In the above, Inequality (24) follows from Oord et al. (2018), which quickly becomes more accurate when $N$ increases. Inequality (27) follows from the data processing inequality (Shwartz-Ziv & Tishby, 2017). Finally, we have $\mathbb{E}[\mathcal{L}_{\mathrm{contrast}}] \geq \log(N-1) - I(h, y)$, and thus minimizing $\mathcal{L}_{\mathrm{contrast}}$ under the stochastic gradient descent framework maximizes a lower bound of $I(h, y)$.

## E  ARCHITECTURE OF AUXILIARY NETWORKS

Here, we introduce the network architectures of $w$, $\psi$ and $\phi$ we use in our experiments. Note that, $w$ is a decoder that aims to reconstruct the input images from deep features, while $\psi$ and $\phi$ share the same architecture except for the last layer. The architectures used on CIFAR, SVHN and STL-10 are shown in Table 7 and Table 8. Architectures on ImageNet are shown in Table 9 and Table 10. The architecture of $\psi$ for the semantic segmentation experiments on Cityscapes is shown in Table 11, where we use the same decoder $w$ as on ImageNet (except for the size of feature maps). An empirical study on the size and architecture of auxiliary nets is presented in Appendix H.

## F  DETAILS OF EXPERIMENTS

**Datasets.** (1) The CIFAR-10 (Krizhevsky et al., 2009) dataset consists of 60,000 32x32 colored images of 10 classes, 50,000 for training and 10,000 for test. We normalize the images with channel

Table 7: Architecture of the decoder $w$ on CIFAR, SVHN and STL-10.

| Input: 32×32 / 16×16 / 8×8 feature maps (96×96 / 48×48 / 24×24 on STL-10) |
|---|
| Bilinear Interpolation to 32×32 (96×96 on STL-10) |
| 3×3 conv., stride=1, padding=1, output channels=12, BatchNorm+ReLU |
| 3×3 conv., stride=1, padding=1, output channels=3, Sigmoid |

Table 8: Architecture of $\psi$ and $\phi$ on CIFAR, SVHN and STL-10.

| Input: 32×32 / 16×16 / 8×8 feature maps (96×96 / 48×48 / 24×24 on STL-10) |
|---|
| 32×32 (96×96) input features: 3×3 conv., stride=2, padding=1, output channels=32, BatchNorm+ReLU |
| 16×16 (48×48) input features: 3×3 conv., stride=2, padding=1, output channels=64, BatchNorm+ReLU |
| 8×8 (24×24) input features: 3×3 conv., stride=1, padding=1, output channels=64, BatchNorm+ReLU |
| Global average pooling |
| Fully connected 32 / 64→128, ReLU |
| Fully connected 128→10 for $\psi$ or 128→128 for $\phi$ |

Table 9: Architecture of the decoder $w$ on ImageNet.

| Input: 28×28 feature maps |
|---|
| 1×1 conv., stride=1, padding=0, output channels=128, BatchNorm+ReLU |
| Bilinear Interpolation to 56×56 |
| 3×3 conv., stride=1, padding=1, output channels=32, BatchNorm+ReLU |
| Bilinear Interpolation to 112×112 |
| 3×3 conv., stride=1, padding=1, output channels=12, BatchNorm+ReLU |
| Bilinear Interpolation to 224×224 |
| 3×3 conv., stride=1, padding=1, output channels=3, Sigmoid |

Table 10: Architecture of $\psi$ on ImageNet.

| Input: 28×28 feature maps |
|---|
| 1×1 conv., stride=1, padding=0, output channels=128, BatchNorm+ReLU |
| 3×3 conv., stride=2, padding=1, output channels=256, BatchNorm+ReLU |
| 3×3 conv., stride=2, padding=1, output channels=512, BatchNorm+ReLU |
| 1×1 conv., stride=1, padding=0, output channels=2048, BatchNorm+ReLU |
| Global average pooling |
| Fully connected 2048→1000 |

Table 11: Architecture of $\psi$ on Cityscapes.

| Input: 64×128 feature maps, 1024 channels |
|---|
| 3×3 conv., stride=1, padding=1, output channels=512, BatchNorm+ReLU |
| Dropout, p=0.1 |
| 1×1 conv., stride=1, padding=0, output channels=19 |

means and standard deviations for pre-processing. Then data augmentation is performed by 4x4 random translation followed by random horizontal flip (He et al., 2016; Huang et al., 2019). (2) SVHN (Netzer et al., 2011) consists of 32x32 colored images of digits. 73,257 images for training and 26,032 images for test are provided. Following Tarvainen & Valpola (2017); Wang et al. (2020a), we perform random 2x2 translation to augment the training set. (3) STL-10 (Coates et al., 2011) contains 5,000 training examples divided into 10 predefined folds with 1000 examples each, and 8,000 images for test. We use all the labeled images for training and test the performance on the provided test set. Data augmentation is performed by 4x4 random translation followed by random horizontal flip. (4) ImageNet is a 1,000-class dataset from ILSVRC2012 (Deng et al., 2009), with 1.2 million images for training and 50,000 images for validation. We adopt the same data augmentation and pre-processing configurations as Huang et al. (2019; 2018b); Wang et al. (2019; 2020b); Yang et al. (2020). (5) Cityscapes dataset (Cordts et al., 2016) contains 5,000 1024×2048 pixel-level finely annotated images (2,975/500/1,525 for training, validation and testing) and 20,000 coarsely annotated images from 50 different cities. Each pixel of the image is categorized among 19 classes.

Following Chen et al. (2017), we conduct our experiments on the finely annotated dataset and report the performance on the validation set. The training images are augmented by randomly scaling (from 0.5 to 2.0) followed by randomly cropping high-resolution patches ($512 \times 1024$ or $640 \times 1280$). At test time, we simply feed the whole $1024 \times 2048$ images into the model.

**Networks and training hyper-parameters.** Our experiments on CIAFR-10, SVHN and STL-10 are based on three popular networks, namely ResNet-32/110 (He et al., 2016) and DenseNet-BC-100-12 (Huang et al., 2019). The networks are trained using a SGD optimizer with a Nesterov momentum of 0.9 for 160 epochs. The L2 weight decay ratio is set to 1e-4. For ResNets, the batch size is set to 1024 and 128 for CIFAR-10/SVHN and STL-10, associated with an initial learning rate of 0.8 and 0.1, respectively. For DenseNets, we use a batch size of 256 and an initial learning rate of 0.2. The cosine learning rate annealing is adopted. Note that, the results of greedy supervised learning presented in Table 1 follows exactly the same experimental configurations stated here. On ImageNet, we train ResNet-101 and ResNet-152 with a batch size of 1024 and an initial learning rate of 0.4. For ResNeXt-101, $32 \times 8$d, we use a batch size of 512 and an initial learning rate of 0.2. The number of training epochs is set to 90. Other hyper-parameters are the same as CIAFR-10. For the DeepLab-V3 model used in semantic segmentation, we follow the training configurations of MMSegmentation (Contributors, 2020) (with ResNet-101 and synthetic batch normalization), except for using an initial learning rate of 0.015 when setting the batch size to 12 or using $640 \times 1280$ cropped patches.

**Local module splitting.** Since ResNets consist of a cascade of residual blocks, which naturally should not be further divided into smaller parts, we view each residual block as a minimal indivisible unit, or say, a basic layer (distinguished from a single convolutional layer). Particularly, the first convolutional layer of the network is individually viewed as a basic layer. As a consequence, ResNet-32 has 16 basic layers, and ResNet-110 has 55 basic layers. If ResNet-32 is split into $K = 8$ local modules, then each module will have 2 basic layers. In the cases where the number of basic layers is not divisible by $K$, we assign one less basic layer to earlier modules. For example, if ResNet-110 is split into $K = 4$ modules, the corresponding numbers of basic layers will be $\{13, 14, 14, 14\}$. If ResNet-110 is split into $K = 16$ modules, the corresponding numbers of basic layers will be $\{3\} \times 9$ modules $+ \{4\} \times 7$ modules. For DenseNet-BC, similarly, we view each dense layer (the composite function BN-ReLU-$1 \times 1$Conv-BN-ReLU-$3 \times 3$Conv) as a basic layer following their paper (Huang et al., 2019). The first convolutional layer and the transition layers are viewed as individual basic layers. The splitting criteria is the same as ResNets. Notably, for InfoPro*, the networks are divided to make sure that each local module has the same memory consumption during training, and hence the aforementioned splitting criteria based on the same numbers of basic layers is not applicable, as we discussed in Section 4.1.

# G  DETAILS OF MUTUAL INFORMATION ESTIMATION

In this section, we describe the details on obtaining the estimates of $I(\boldsymbol{h}, \boldsymbol{x})$ and $I(\boldsymbol{h}, y)$ we present in Figures 2, 3 and 6.

As we have discussed in Section 3.3, the expected reconstruction error $\mathcal{R}(\boldsymbol{x}|\boldsymbol{h})$ follows $I(\boldsymbol{h}, \boldsymbol{x}) = H(\boldsymbol{x}) - H(\boldsymbol{x}|\boldsymbol{h}) \geq H(\boldsymbol{x}) - \mathcal{R}(\boldsymbol{x}|\boldsymbol{h})$ (Vincent et al., 2008; Rifai et al., 2012; Kingma & Welling, 2013; Makhzani et al., 2015; Hjelm et al., 2019). Therefore, similar to Section 3.3, we estimate $I(\boldsymbol{h}, \boldsymbol{x})$ by training a decoder parameterized by $\boldsymbol{w}$ to obtain the minimal reconstruction loss, namely $I(\boldsymbol{h}, \boldsymbol{x}) \approx \max_{\boldsymbol{w}}[H(\boldsymbol{x}) - \mathcal{R}_{\boldsymbol{w}}(\boldsymbol{x}|\boldsymbol{h})]$. Note that, ideally, this bound can be arbitrarily tight provided that $\boldsymbol{w}$ has sufficient capacity. In specific, we use the same network architecture as Table 7, and train it for 10 epochs to minimize the averaged binary cross-entropy reconstruction loss of each pixel. An adam (Kingma & Ba, 2014) optimizer with default hyper-parameters (lr=0.001, betas=(0.9, 0.999), eps=1e-08, weight_decay=0) is adopted. Naive as this procedure might seems, for one thing, we find that it is sufficient to reconstruct the input images well given enough information, and meanwhile distinguish different values of $I(\boldsymbol{h}, \boldsymbol{x})$ via the quality of reconstructed images, as shown in Figure 7. For another, we are primarily concerned with the comparisons of $I(\boldsymbol{h}, \boldsymbol{x})$ between end-to-end training and various cases of greedy supervised learning rather than obtaining the exact values of $I(\boldsymbol{h}, \boldsymbol{x})$. The same training process is applied to all the intermediate features $\boldsymbol{h}$, and hence the comparisons are fair. Finally, since $H(\boldsymbol{x})$ is a constant, for the ease of understanding, we simply present $1 - AverageBinaryCrossEntropyLoss(\boldsymbol{x}|\boldsymbol{h})$ as the estimates of $I(\boldsymbol{h}, \boldsymbol{x})$, equivalent to adding the constant $1 - H(\boldsymbol{x})$ to the real estimates of $I(\boldsymbol{h}, \boldsymbol{x})$.

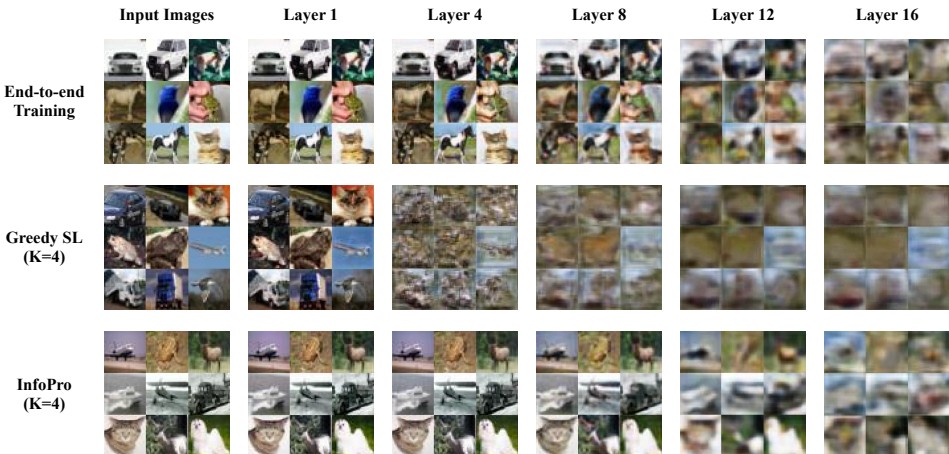

Figure 7: Visualization of the reconstruction results obtained from the decoder $\boldsymbol{w}$.

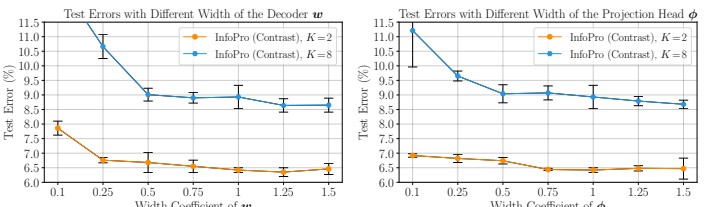

Figure 8: Test errors of ResNet-110 trained by InfoPro (Contrast) on CIFAR-10, with varying width of the decoder $\boldsymbol{w}$ (*left*) and the projection head $\boldsymbol{\phi}$ (*right*).

Table 12: Test errors of ResNet-110 trained by InfoPro (Contrast) on CIFAR-10, with different architecture of $\boldsymbol{\phi}$. "MLP" refers to the multi-layer perceptron.

| Architecture of $\phi$ | $K=2$ | $K=8$ |
|---|---|---|
| 2-MLP | $7.19 \pm 0.22\%$ | $10.63 \pm 0.45\%$ |
| 1 Conv + 1 Linear | $6.84 \pm 0.05\%$ | $9.80 \pm 0.26\%$ |
| 1 Conv + 2-MLP (*ours*) | $6.42 \pm 0.08\%$ | $8.93 \pm 0.40\%$ |
| 1 Conv + 3-MLP | $6.40 \pm 0.09\%$ | $8.79 \pm 0.39\%$ |
| 2 Conv + 2-MLP | $6.07 \pm 0.17\%$ | $7.87 \pm 0.99\%$ |

For $I(\boldsymbol{h}, y)$, as we have discussed in Section 3.3 as well, since $I(\boldsymbol{h}, y) = H(y) - H(y|\boldsymbol{h}) = H(y) - \mathbb{E}_{(\boldsymbol{h}, y)}[-\log p(y|\boldsymbol{h})]$, we train an auxiliary classifier $q_{\boldsymbol{\psi}}(y|\boldsymbol{h})$ with parameters $\boldsymbol{\psi}$ to approximate $p(y|\boldsymbol{h})$, such that we have $I(\boldsymbol{h}, y) \approx \max_{\boldsymbol{\psi}}\{H(y) - \frac{1}{N}[\sum_{i=1}^{N} -\log q_{\boldsymbol{\psi}}(y_i|\boldsymbol{h}_i)]\}$. Here we simply adopt the test accuracy of $q_{\boldsymbol{\psi}}(y|\boldsymbol{h})$ as the estimate of $I(\boldsymbol{h}, y)$, which is highly correlated to the value of $-\frac{1}{N}[\sum_{i=1}^{N} -\log q_{\boldsymbol{\psi}}(y_i|\boldsymbol{h}_i)]\}$ (or say, the cross-entropy loss). This can be viewed as the highest generation performance that a classifier based on $\boldsymbol{h}$ is able to reach. Notably, we use a ResNet-32 as $q_{\boldsymbol{\psi}}$. For the inputs of $q_{\boldsymbol{\psi}}$, we up-sample $\boldsymbol{h}$ to $32 \times 32$ and map $\boldsymbol{h}$ to 16 channels at the first layer. All training hyper-parameters of the ResNet-32 are the same as Appendix F.

# H  MORE RESULTS

**Size and architecture of auxiliary nets**. Here we investigate how the auxiliary nets (i.e., $\boldsymbol{w}$, $\boldsymbol{\psi}$ and $\boldsymbol{\phi}$) influence the performance of our method. Since $\boldsymbol{\psi}$ and $\boldsymbol{\phi}$ share the same architecture, we study the decoder $\boldsymbol{w}$ and the projection head $\boldsymbol{\phi}$ for example. We start by scaling their width with a certain factor (which equals 1 for our original design), and show the results in Figure 8. It can be observed that using larger $\boldsymbol{w}$ and $\boldsymbol{\phi}$ both improve the performance, but the effects are less significant. In addition, their sizes can be shrunk by up to 2 times without severely degrading the accuracy. We also test other architectures for $\boldsymbol{\phi}$ in Table 12, where "Conv" and "Linear" refer to convolutional and linear layers, respectively. We find that involving at least one conv layer and a following MLP instead of a simple linear layer are both important designs. Besides, although adding more conv layers further boosts the performance, we observe that this comes at a considerably increased computational overhead.

**Temperature** $\tau$. In Figure 9, we change the temperature $\tau$ for InfoPro (Contrast). Our method is robust to $\tau$ when $\tau \leq 0.1$. However, we find the training tends to be unstable when $\tau < 0.005$.

**Batch size**. For a comprehensive comparison of InfoPro (Softmax) and InfoPro (Contrast), we vary the batch sizes for the SGD optimizer, and present the results in Table 13. It is shown that training models with small mini-batches for sufficient epochs is beneficial for InfoPro (Softmax), but produces limited positive effects on InfoPro (Contrast). We also note that using small mini-batches

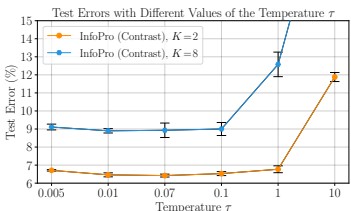

Figure 9: Performance of InfoPro (Contrast) with varying temperature $\tau$. Test errors of ResNet-110 on CIFAR-10.

Table 13: Performance of InfoPro (Contrast/Softmax) with varying batch sizes. Two settings are considered: training models with the same number of iterations (40/80/160 epochs for batch size=256/512/1024) and epochs (160 epochs). All other training hyper-parameters (i.e., the learning rate schedule, weight decay, etc.) are remained unchanged. Test errors of ResNet-110 on CIFAR-10 are reported.

| Training Epochs | | 40 | 80 | 160 | | |
|---|---|---|---|---|---|---|
| Batch Size | | 256 | 512 | 1024 | 512 | 256 |
| $K=2$ | InfoPro (Softmax) | $\mathbf{8.88 \pm 0.36\%}$ | $7.70 \pm 0.23\%$ | $7.01 \pm 0.34\%$ | $\mathbf{6.14 \pm 0.11\%}$ | $\mathbf{5.95 \pm 0.19\%}$ |
| | InfoPro (Contrast) | $9.96 \pm 0.29\%$ | $\mathbf{7.49 \pm 0.35\%}$ | $\mathbf{6.42 \pm 0.08\%}$ | $6.16 \pm 0.15\%$ | $6.19 \pm 0.20\%$ |
| $K=8$ | InfoPro (Softmax) | $\mathbf{11.22 \pm 0.10\%}$ | $\mathbf{9.42 \pm 0.05\%}$ | $9.40 \pm 0.27\%$ | $\mathbf{8.37 \pm 0.33\%}$ | $\mathbf{8.04 \pm 0.29\%}$ |
| | InfoPro (Contrast) | $13.22 \pm 0.57\%$ | $9.96 \pm 0.13\%$ | $\mathbf{8.93 \pm 0.40\%}$ | $9.02 \pm 1.18\%$ | $9.32 \pm 1.44\%$ |

Table 14: Performance of InfoPro with the VGG network (Simonyan & Zisserman, 2014). The averaged test errors and standard deviations of 5 independent trials are reported. InfoPro (Softmax/Contrast) refers to two approaches to estimating $I(\boldsymbol{h}, y)$.

| Dataset | Network | Method | $K = 2$ | $K = 4$ |
|---|---|---|---|---|
| CIFAR-10 | VGG-11 (w/ BatchNorm) (E2E: $8.10 \pm 0.14\%$) | Greedy SL | $8.88 \pm 0.11\%$ | $10.04 \pm 0.31\%$ |
| | | DGL (Belilovsky et al., 2020) | $8.25 \pm 0.12\%$ | $8.46 \pm 0.29\%$ |
| | | InfoPro (Softmax) | $8.30 \pm 0.10\%$ | $8.36 \pm 0.16\%$ |
| | | InfoPro (Contrast) | $\mathbf{8.18 \pm 0.15\%}$ | $\mathbf{8.23 \pm 0.19\%}$ |

Table 15: Object detection results on COCO (Lin et al., 2014). We initialize the backbone of Faster-RCNN-FPN (Ren et al., 2015) using ResNet-101 trained by E2E training and InfoPro*, $K = 2$ on ImageNet. The COCO style box average precision (AP) metric is adopted, where $AP_{50}$ and $AP_{75}$ denote AP over $50\%$ and $75\%$ IoU thresholds, mAP takes the average value of AP over different thresholds ($50\%$–$95\%$), and $mAP_S$, $mAP_M$ and $mAP_L$ denote mAP for objects at different scales. All results are presented in percentages (%). The better results are **bold-faced**.

| Method | Backbone | mAP | $AP_{50}$ | $AP_{75}$ | $mAP_S$ | $mAP_M$ | $mAP_L$ |
|---|---|---|---|---|---|---|---|
| Faster-RCNN-FPN (Ren et al., 2015) | ResNet-101 (E2E training) | 40.0 | 60.7 | 43.7 | 22.7 | 44.2 | 51.4 |
| | ResNet-101 (InfoPro*, $K = 2$) | **40.2** | **61.1** | **44.0** | **23.4** | **44.4** | **52.1** |

usually prolongs the practical training time as they cannot make full use of even a single GPU (Nvidia Titan Xp). Besides, when considering a short schedule with the same number of updates (iterations), adopting small mini-batches significantly hurts the performance of InfoPro (Contrast). This observation is consistent with previous works (Chen et al., 2020; He et al., 2020; Khosla et al., 2020).

**Empirical study on the tightness of the upper bound.** Here, we empirically study the gap between of the upper bound $\bar{\mathcal{L}}_{\text{InfoPro}}$ and $\mathcal{L}_{\text{InfoPro}}$. Since the gap $\epsilon = \bar{\mathcal{L}}_{\text{InfoPro}} - \mathcal{L}_{\text{InfoPro}}$ is upper bounded by $\epsilon \leq \lambda_2 \left[ I(\boldsymbol{x}, y) - I(\boldsymbol{h}, y) \right]$, we simply need to check the gap between $I(\boldsymbol{x}, y)$ and $I(\boldsymbol{h}, y)$. To this end, we consider training a ResNet-110 on CIFAR-10 using InfoPro (Contrast) with $K = 2$, and estimate the mutual information $I(\boldsymbol{h}, y)$ between the outputs of the first local module and the label. In addition, to obtain a comparable estimate of $I(\boldsymbol{x}, y)$, we assume that in end-to-end learned networks, the intermediate features retain all the task-relevant information within the original inputs (which is empirically

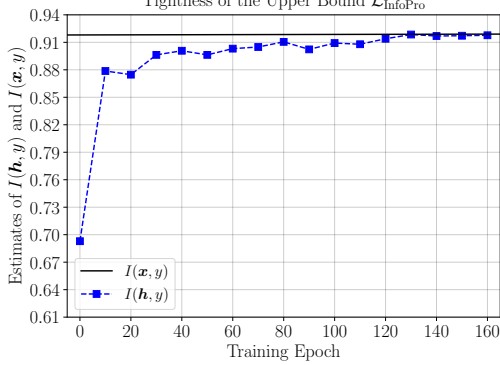

Figure 10: The estimates of $I(\boldsymbol{x}, y)$ and $I(\boldsymbol{h}, y)$.

observed in this paper). Hence, we can estimate $I(\boldsymbol{x}, y)$ by training the network in an end-to-end fashion, and estimating the mutual information $I(\boldsymbol{h}, y)$ in the same position as the end of the first module. The comparisons between the estimates of $I(\boldsymbol{x}, y)$ and $I(\boldsymbol{h}, y)$ are presented in Figure 10. One can observe that the gap $\epsilon$ shrinks gradually during the training process.

**Results with VGG** are presented in Table 14. Obviously, similar observations to Table 2 can be obtained. InfoPro achieves competitive performance with E2E training, while outperforming DGL.

**Transferability.** To verify the transferability of the models trained by the proposed method, we initialize the backbone of Faster-RCNN (Ren et al., 2015) using ResNet-101 trained by E2E training

and InfoPro*, $K = 2$ on ImageNet, and train it for the MS COCO (Lin et al., 2014) object detection task. The training process adopts the default configuration of MMDetection (Chen et al., 2019) with feature pyramid networks (FPN) (Lin et al., 2017). The results are reported in Table 15. It can be observed that the locally learned backbone using InfoPro* outperforms its E2E counterpart in terms of average precision (AP). This result is consistent with the accuracy on ImageNet.

# I    ADDITIONAL DISCUSSIONS AND FUTURE WORK

**Applying InfoPro to regression tasks.** Although this paper mainly focuses on implementing InfoPro in the context of classification based tasks (i.e., image classification and semantic segmentation), the formulation of $\mathcal{L}_{\text{InfoPro}}$ and $\bar{\mathcal{L}}_{\text{InfoPro}}$ is general and flexible. As long as the mutual information $I(\boldsymbol{h}, \boldsymbol{x})$ and $I(\boldsymbol{h}, y)$ can be estimated, InfoPro is able to be used in more tasks. In vision tasks, $I(\boldsymbol{h}, \boldsymbol{x})$ can usually be estimated with a decoder, as we introduced in Section 3.3, while for estimating $I(\boldsymbol{h}, y)$, the technique we discussed may be easily extended to the regression tasks (e.g., depth estimation, bounding box regression in object detection). For example, consider a target value $y_i \in [0, 1]$ corresponding to the sample $\boldsymbol{x}_i$ and the hidden representation $\boldsymbol{h}_i$. It can be viewed as a Bernoulli distribution where $\text{P}(y = 1) = y_i$, such that we have $I(\boldsymbol{h}, y) \approx \max_{\boldsymbol{\psi}} \{ H(y) - \frac{1}{N} [\sum_{i=1}^{N} -y_i \log q_{\boldsymbol{\psi}}(y = 1|\boldsymbol{h}_i) - (1 - y_i) \log q_{\boldsymbol{\psi}}(y = 0|\boldsymbol{h}_i)] \}$. As a result, the auxiliary network $q_{\boldsymbol{\psi}}(y|\boldsymbol{h})$ can be trained with the binary cross-entropy loss. This might also be approximated by the mean-square loss as it has the same minima as the binary cross-entropy loss. In the future, we will focus on applying InfoPro to more complex tasks, such as 2D/3D detection, instance segmentation and video recognition.

