# OpenReview forum: "Revisiting Locally Supervised Learning: an Alternative to End-to-end Training"
_ICLR.cc/2021/Conference — ICLR 2021 Poster_

### Official Review · AnonReviewer4 · 2020-10-25
**Good motivation but lack of comparisons**

**Rating:** 6
**Confidence:** 4

**Review:**

This paper analyzed the reason why locally supervised training led to performance degradation. And based on the analysis, the author proposed the information propagation loss (can be understood as the combination of a classification loss and a reconstruction loss) aiming to prevent information collapse. Equipped with the proposed method, 40% memory footprint can be reduced demonstrated by their experiments, which is surprising.

Strength
(1) Deep neural networks need heavy memory cost, especially for object detection and segmentation task. Reducing memory cost in the training can enlarge the batch size in the training, thus speed up the training process. This is a very promising direction.
(2) The paper is well motivated by analyzing traditional locally supervised training. Moreover, the analyze of information in features is also informative and reasonable.
(3) The experimental results are good. The proposed method can achieve comparable performance with 40% less memory footprint.
(4) The paper is clearly and is easy to understand.
(5) The proposed method is simple but efficient. And it can be used in different tasks, like classification and semantic segmentation.

Weaknesses
(1) This is seldom comparisons with other methods in this paper. Only comparisons with the baseline are provided.
(2) Is the computational overhead is measured by training time or theorical computation cost? The comparison of real training time is more informative.
(3) Comparisons with other methods for reducing the memory consumption, e.g., [a] should also be added since they share similar aim. [a] Chen et al. Training Deep Nets with Sublinear Memory Cost.
(4) Recent contrastive learning works show that hyper-parameter τ in contrastive loss plays a very important role, like MOCO. Though τ = 0.07 is fit for MOCO, the setting here is different from MOCO. In this paper, the contrastive loss makes use of label information. If the ablation study for training hyper-parameter τ in the contrastive loss function is provided, it will be good.
(5) In the experiments, the contrastive loss outperforms cross-entropy loss. And the authors attribute this phenomenon to using larger batch size for contrastive loss. However, I think the strong regularization on intra-class representation may be the key factor. Anyhow, ablation study of different batch sizes for contrastive loss should be added here.

---

> ### Author Response · Authors · 2020-11-20
> **Response to Reviewer 4**
>
> Thank you for your valuable comments on the experiments. We have added the results with regards to your concerns. Our responses are listed below:
>
>
> **Q1**: This is seldom comparisons with other methods in this paper. Only comparisons with the baseline are provided.
> **A1**: Thank you for the suggestion. The results of the state-of-the-art baseline, DGL (published in ICML 2020), have been added in Table 2. Our method compares favorably with it in most of the settings.
>
>
> **Q2**: Is the computational overhead is measured by training time or theorical computation cost? The comparison of real training time is more informative.
> **A2**: The computational overhead reported in the original paper is the theoretical result. In the revised paper, we have added the practical additional training time (wall time cost), and clearly distinguished the theoretical and practical cost (see: Tables 3, 4, 5).
>
> **Q3**: Comparisons with other methods for reducing the memory consumption, e.g., [a] should also be added since they share similar aim. [a] Chen et al. Training Deep Nets with Sublinear Memory Cost.
> **A3**: In our revision, we compare our method with this baseline in Table 3. Our method achieves competitive performance with it, but significantly reduces the computational and time cost.
>
> **Q4**: Recent contrastive learning works show that hyper-parameter τ in contrastive loss plays a very important role, like MOCO. Though τ = 0.07 is fit for MOCO, the setting here is different from MOCO. In this paper, the contrastive loss makes use of label information. If the ablation study for training hyper-parameter τ in the contrastive loss function is provided, it will be good.
> **A4**: As suggested, we have conducted the sensitivity test for the temperature $\tau$ and reported the results in Figure 5. InfoPro (Contrast) is relatively robust to $\tau$ when $0.005 < \tau < 0.1$.
>
>
> **Q5**: In the experiments, the contrastive loss outperforms cross-entropy loss. And the authors attribute this phenomenon to using larger batch size for contrastive loss. However, I think the strong regularization on intra-class representation may be the key factor. Anyhow, ablation study of different batch sizes for contrastive loss should be added here.
> **A5**: Thank you for the suggestion. We have added the results of the ablation study for different batch sizes in Table 7. We find that the contrastive loss generally benefits from a large batch size and a long training schedule. This observation is consistent with existing work [1]. In addition, we totally agree that the regularization on intra-class representation is an important factor of its strong performance. As a matter of fact, we believe that with larger mini-batches, where there will be more samples in each class, both the intra-class and the inter-class regularization effects will grow stronger, leading to better performance.
> [1] Ting Chen, Simon Kornblith, Mohammad Norouzi, and Geoffrey Hinton. A simple framework for contrastive learning of visual representations. In ICML, 2020.

---

### Official Review · AnonReviewer2 · 2020-10-27
**Insightful, some shortcomings in empirical evaluation**

**Rating:** 6
**Confidence:** 3

**Review:**

The paper proposes a strategy for training feed-forward networks in a more memory-efficient manner by employing local as opposed to end-to-end supervision. End-to-end/global (E2E) supervision as the dominant paradigm in training deep networks considers a loss function at the very end of the network for backpropagation of the resulting gradients, whereas local supervision injects supervisory signals (such as the same E2E objective, e.g. classification) at intermediate layers in the network. The benefit of such intermediate supervision is the ability to train larger networks in smaller chunks piece by piece, where each individual training is more memory efficient due to reduced need to store activations (and weights and biases) in GPU memory. As a drawback, however, it had been shown earlier that such local training is less optimal than global training in terms of the achievable generalization performance. The authors propose a new training strategy that aims at combining the memory efficiency of local supervision and piecewise training with the error performance of global training. Considering a given intermediate layer, the paper motivates to maximize the mutual information between the activations in this layer and the input signal to retain relevant information, while minimizing the mutual information of the activations and a nuisance variable, where the nuisance is defined as having no mutual information with the target variable (e.g. the classification prediction). The authors argue that this local supervision allows to train the features at the intermediate layer such that they carry relevant information from the input to the target variable without resorting to direct supervision with the target variable. Direct computation of the nuisance variable is infeasible and the authors propose a bounded approximation.
Empirical results are discussed on five common vision datasets and two CNN architectures in relation to the existing state of the art.

### Strengths
**[S1]** The paper is largely written well and addresses a relevant problem. Contributions and claims are laid out well.

**[S2]** The method has potentially multiple benefits: (a) more memory efficient training without loss of performance, (b) speedups for asynchronous training, (c) use of larger batch size or larger models.

**[S3]** The method is motivated reasonably well by the argument of minimizing the loss of mutual information with the input variable while minimizing the mutual information of nuisance variables with respect to the target quantity.

**[S4]** The inclusion of Greedy SL+ to account for the additional degrees of freedom of the proxy networks in InfoPro is a good attention to detail in the empirical evaluation.

**[S5]** The paper looks at a reasonable set of tasks and studies the performance on relevant data, for instance Imagenet and Cityscapes semantic segmentation (but consider [W1] below)

### Weaknesses:
**[W1]** Baselines of empirical comparisons: With the exception of Fig 3, the empirical results are compared to only a comparatively simple baseline (Greedy SL/SL+), but not the same state of the art mentioned in section 4.1 (DGL, DIB, BoostResNet). This seems a rather odd omission, particularly as Fig 3/Section 4.1 elaborate on alternative methods (DGL, DIB, BoostResNet). It would support the paper's case to either include the results of those in table 2 or elaborate on their absence.

**[W2]** Fig 3: The scale of the y-axis is chosen in a way that a casual reader may visually misinterpret the absolute difference in error of the shown methods. For instance, at first glance DGL at K=2 seems twice as bad as Infopro, while in actuality is only about 14% worse (error from 7.76% to ~8.8%)

**[W3]** A methodical concern is the choice of $\phi$ and $\psi$ and the limited discussion around their choice (section 3.3, App. E): What is the sensitivity of the optimization with respect to their size and structure, what happens if I make them larger or smaller, how small can I make them? How would they look like for objectives other than classification?

### Further comments:
**[C1]** It would be insightful to consider what the implications for networks with recurrent structures would be.

**[C2]** It took a while to infer in section 3.3. that $\mathcal{R}$ is the reconstruction of the input data. A short sentence there may help future readers go through more smoothly.

I feel that I have learned something from the paper. It discusses the contributions, motivations and method reasonably thorough. My concerns are largely around the empirical substantiation of the claims, see [W1].

---

> ### Author Response · Authors · 2020-11-20
> **Response to Reviewer 2**
>
> Thank you for your valuable suggestions on the empirical evaluation. We have revised the paper by adding the results with regards to your concerns. Our responses are listed below:
>
>
> **Q1**: [W1] Baselines of empirical comparisons: With the exception of Fig 3, the empirical results are compared to only a comparatively simple baseline (Greedy SL/SL+), but not the same state of the art mentioned in section 4.1 (DGL, DIB, BoostResNet). This seems a rather odd omission, particularly as Fig 3/Section 4.1 elaborate on alternative methods (DGL, DIB, BoostResNet). It would support the paper's case to either include the results of those in table 2 or elaborate on their absence.
> **A1**: We have added the results of the state-of-the-art baseline, DGL (published in ICML 2020), in Table 2. Our method compares favorably with it. In addition, DIB and BoostResNet are both layer-wise training methods, while our method performs layer-wise training only when it is applied to ResNet-32 (K=16). For a fair comparison, we present the results of DIB and BoostResNet in Figure 3 instead of Table 2.
>
> **Q2**: [W2] Fig 3: The scale of the y-axis is chosen in a way that a casual reader may visually misinterpret the absolute difference in error of the shown methods. For instance, at first glance DGL at K=2 seems twice as bad as Infopro, while in actuality is only about 14% worse (error from 7.76% to ~8.8%).
> **A2**: Thank for the suggestion. We have revised Figure 3 by resizing the y-axis to make it start from 0.
>
> **Q3**: [W3] A methodical concern is the choice of $\phi$ and $\psi$ and the limited discussion around their choice (section 3.3, App. E): What is the sensitivity of the optimization with respect to their size and structure, what happens if I make them larger or smaller, how small can I make them? How would they look like for objectives other than classification?
> **A3**: (1) Thank you for the valuable comment. We have added a detailed analysis on both the size and architecture of the auxiliary networks (see: Figure 4 and Table 6). In specific, since $\phi$ and $\psi$ have the same network architecture except only for the final linear layer, we study $\phi$ for example. We first scale its width with a certain factor (which equals 1 for the original design), and show that the size of $\phi$ can be shrunk by up to 2 times without severely degrading the accuracy. On the other hand, using a wider $\phi$ makes little sense in terms of the performance. We also test other architectures for $\phi$, including inserting/removing convolutional layers and linear lavers. It shows that involving at least 1 convolutional layer in $\phi$ is important. For more details, please refer to the revised paper.
> (2) The auxiliary network $\phi$ or $\psi$ is used to estimate the task-relevant information $I(h,y)$. Apart from classification tasks, the technique for estimating $I(h,y)$ we mentioned in section 3.3 can be easily extended to the regression tasks (e.g., depth estimation, bounding box regression for object detection). For example, consider a target value $y_i \in [0, 1]$ corresponding to the sample $x_i $ and the hidden representation $h_i$. It can be view as a Bernoulli distribution where $P(y=1)=y_i$. As a result, the auxiliary network $q_{\psi}(y|{h})$ can be trained with the binary cross-entropy loss. This might also be approximated by the mean-square loss as it has the same minima as the binary cross-entropy loss. We have added more discussions in Appendix J, please refer to it for more details.
>
> **Q4**: [C1] It would be insightful to consider what the implications for networks with recurrent structures would be.
> **A4**: Thank you for pointing out the promising direction. However, it may not be straightforward to directly transfer InfoPro to recurrent neural networks (RNN). Consider inputting a sentence to RNN, if we split RNN in the temporal dimension, the sentence will also be split to short sub-sentences. Each sub-sentence may not contain enough information with regard to the target by itself. A possible direction might be implementing InfoPro in stacked recurrent networks, where we may consider training earlier and later recurrent units isolatedly with local supervision. We will focus on this in our future work.
>
> **Q5**: [C2] It took a while to infer in section 3.3. that $\mathcal{R}$ is the reconstruction of the input data. A short sentence there may help future readers go through more smoothly.
> **A5**: Thank you for the suggestion. We have revised the paper to make it clear.

---

### Official Review · AnonReviewer3 · 2020-10-28

**Rating:** 7
**Confidence:** 3

**Review:**

# Summary

The paper analyzes the pitfalls of locally supervised learning from the point of view of information propagation and proposes a new auxiliary loss that can facilitate locally supervised learning. The proposed loss, "infopro loss", is then relaxed to a tractable upper bound, which is then used instead. To implement the loss, mutual information is approximated with a decoder, as well as a classifier. The authors further introduce now contrastive learning fits in the framework as a lower bound maximization process regarding mutual information. The experimental results on standard datasets demonstrate the efficacy of the proposed method.

I have enjoyed reading the paper quite a lot, and therefore recommend accepting the paper. Still, I have some reservations that I would love the authors to clarify via the rebuttal.

# Strength

The paper is well written. There are some issues (which I detail in the weaknesses section), but most are very clear and easy to follow. There are some parts that are repetitive, but it does allow readers to scheme through without missing the important point.

The part that I like most about the paper is how proposition 1 and appendix D is presented. They are theoretically well-motivated and gladly seems to work, despite the relaxation. I personally think appendix D deserves more attention in the main text, but this is my personal preference.

The results clearly show that the method improves over simple greedy locally supervised learning, as well as other attempts at this problem.

# Weaknesses

## Section 3.1, regarding equation (1)

I am not 100% sure on this, but shouldn't the third last sentence of Section 3.1 read "...under the goal of retaining as much information of the input as possible"? I am not sure this is actually a constraint, and the I(h,x) term is not explicitly task-relevant information.

## Section 3.3, estimating I(h,y)

It is not clear to me how I(h,y) is finally approximated. Shouldn't p(y|h) disappear after approximation? In the provided approximation it still exists. If p(y|h) is somehow directly used, isn't q not needed at all?

## Section 3.3 final equation

In my opinion, even when it is not referred to in text, equations should have numbers so that future readers can refer to it.

## Regarding Asy-InfoPro

It is somewhat unclear whether the dynamic caching was used at the end. Are the experiments in Table 2 with dynamic caching? From my current understanding, it does not seem to be the case, which leads to my second issue.

This is assuming that the results are without dynamic caching as this seems most logical. The explanation in the "Asynchronous and parallel training" paragraph was not obvious to me during the first read. The second sentence could be rephrased and split so that it becomes clear that the distinction between the two modes is that transient feature maps are seen/unseen and that this has a regularizing effect.  This then brings up the important question, whether the dynamic caching version suffers from the same fate---it should not. Having this experimental verification would greatly strengthen the observations in this paper.

## Softmax vs Contrastive

I am curious as to why contrastive works better. Could it be a coincidence of better hyperparameter tuning? Because the softmax version is a direct estimate on mutual information, whereas the contrastive one optimizes the lower bound.  It would be nice if this was discussed in more detail (I do understand that there is already very little space though!)

## Computational overhead

Is the computational overhead including the memory transfer cost? Are the results the actual physical measures observed via monitoring the resource usage? Or are they theoretical? I might have missed it, but this is not very clear to me.

## Typos and grammar errors

There are quite a few grammatical mistakes throughout the paper. For example, at the beginning of Section 2, it is more natural to write "we start by considering" than "we start with considering". In the italic question in the second paragraph of Section 2, "..., even the former..." should be "..., even though the former...". While I generally did not find these errors to be critical, I would suggest a thorough proofread.

I also found a typo in Appendix B. The last sentence should refer to equation (10) not (9).

## Early stopping, and the choice of the number of epochs

The training process in this paper does not utilize early stopping. While this is somewhat mitigated by the fact that multiple runs are performed, this is in fact another source of overfitting to the dataset and is strictly speaking tuning hyperparameters on the test set. This is a practice that should be avoided.

---

> ### Author Response · Authors · 2020-11-20
> **Response to Reviewer 3 (part 2)**
>
> **Q5**: \# Softmax vs Contrastive \# I am curious as to why contrastive works better. Could it be a coincidence of better hyperparameter tuning? Because the softmax version is a direct estimate on mutual information, whereas the contrastive one optimizes the lower bound. It would be nice if this was discussed in more detail (I do understand that there is already very little space though!)
> **A5**: We have reported more analytical results in the revised paper, including 1) changing the temperature $\tau$ for the contrastive loss (Figure 5), and 2) changing the batch size for training (Table 7). The results suggest that, the higher performance of InfoPro (Contrast) compared with InfoPro (Softmax) comes from adopting a proper temperature $\tau$, a relatively large batch size, and a long training schedule. For one thing, InfoPro (Contrast) introduces an additional tunable hyper-parameter (the temperature $\tau$), which affects the performance. For another, as we show in Appendix D, the lower bound estimated by the contrastive loss tends to be tight with a large batch size, leading to the improved test accuracy. In addition, we find that InforPro needs a relatively long schedule to converge.
>
> **Q6**: \# Computational overhead \# Is the computational overhead including the memory transfer cost? Are the results the actual physical measures observed via monitoring the resource usage? Or are they theoretical? I might have missed it, but this is not very clear to me.
> **A6**: The computational overhead reported in the original paper is the theoretical result. However, in the revised paper, we have added the practical additional training time (wall time cost), and clearly distinguished the theoretical and practical cost (see: Tables 3, 4, 5).
>
> **Q7**: \# Typos and grammar errors \# There are quite a few grammatical mistakes throughout the paper. For example, at the beginning of Section 2, it is more natural to write "we start by considering" than "we start with considering". In the italic question in the second paragraph of Section 2, "..., even the former..." should be "..., even though the former...". While I generally did not find these errors to be critical, I would suggest a thorough proofread. I also found a typo in Appendix B. The last sentence should refer to equation (10) not (9).
> **A7**: Thanks for your comments. We have carefully checked and revised the whole paper for several times according to your suggestion. All typos we found have been corrected. The writing is should be more acceptable now.
>
> **Q8**: \# Early stopping, and the choice of the number of epochs \# The training process in this paper does not utilize early stopping. While this is somewhat mitigated by the fact that multiple runs are performed, this is in fact another source of overfitting to the dataset and is strictly speaking tuning hyperparameters on the test set. This is a practice that should be avoided.
> **A8**: Thank you for the comment. However, we want to clarify that we have never tuned the number of epochs on the test set. We tried our best to avoid (potential) overfitting to the test set. For training epochs, we simply follow existing work [1, 2], and do not change them if not specially mentioned.
> [1] He K, Zhang X, Ren S, et al. Deep residual learning for image recognition[C]//Proceedings of the IEEE conference on computer vision and pattern recognition. 2016: 770-778.
> [2] Huang G, Liu Z, Van Der Maaten L, et al. Densely connected convolutional networks[C]//Proceedings of the IEEE conference on computer vision and pattern recognition. 2017: 4700-4708.

---

> ### Author Response · Authors · 2020-11-20
> **Response to Reviewer 3 (part 1)**
>
> Thank you for your detailed and encouraging comments. We have carefully revised our paper according to the suggestions. Our responses are listed below:
>
> **Q1**: \# Section 3.1, regarding equation (1) \# I am not 100% sure on this, but shouldn't the third last sentence of Section 3.1 read "...under the goal of retaining as much information of the input as possible"? I am not sure this is actually a constraint, and the I(h,x) term is not explicitly task-relevant information.
> **A1**: Indeed, the major effect of minimizing $L_{\text{InfoPro}}(h)$ should be maximally discarding the task-irrelevant information under the goal of retaining as much information of the input as possible. It is not a constraint. We have revised it for a clear description.
>
> **Q2**: \# Section 3.3, estimating I(h,y) \# It is not clear to me how I(h,y) is finally approximated. Shouldn't p(y|h) disappear after approximation? In the provided approximation it still exists. If p(y|h) is somehow directly used, isn't q not needed at all?
> **A2**: Thank you for the valuable comment. In fact, we estimate the expectation over $h$ by sampling $(x_i, h_i, y_i)$ from the true distribution (i.e., $x_i$, $h _i$ and $y_i$ refer to the training sample, the hidden representation and the label respectively). As a result, $p(y|h)$ is only computed as $p(y_i|h_i)$ in the final form, instead of being directly used. We have revised our paper to make it clear. Please refer to Section 3.3 for details.
>
> **Q3**: \# Section 3.3 final equation \# In my opinion, even when it is not referred to in text, equations should have numbers so that future readers can refer to it
> **A3**: Thank you for the comment. We have added a number to this equation.
>
> **Q4**: \# Regarding Asy-InfoPro 1/2 \# It is somewhat unclear whether the dynamic caching was used at the end. Are the experiments in Table 2 with dynamic caching? From my current understanding, it does not seem to be the case, which leads to my second issue.
> \# Regarding Asy-InfoPro 2/2 \# This is assuming that the results are without dynamic caching as this seems most logical. The explanation in the "Asynchronous and parallel training" paragraph was not obvious to me during the first read. The second sentence could be rephrased and split so that it becomes clear that the distinction between the two modes is that transient feature maps are seen/unseen and that this has a regularizing effect. This then brings up the important question, whether the dynamic caching version suffers from the same fate---it should not. Having this experimental verification would greatly strengthen the observations in this paper.
> **A4**: (1) Indeed, the results presented in Table 2 do not use the dynamic caching technique. We have revised Section 4 (both the "Two training modes" paragraph the "Asynchronous and parallel training" paragraph) to make it clearer.
> (2) The dynamic caching version of InfoPro inherently does not suffer from lacking regularizing effects from the noisy outputs of earlier modules during training. Because all its training process is exactly the same as simultaneous training (i.e., “InfoPro (Contrast/Softmax)” in Table 2). The only difference is that we dynamically cache the outputs of earlier modules and use them to train later modules on another GPU. We have revised the "Asynchronous and parallel training" paragraph to highlight this point.

---

### Official Review · AnonReviewer1 · 2020-10-28
**Reviewer1 Comments**

**Rating:** 7
**Confidence:** 4

**Review:**

Briefing:
This paper proposes a new infoPro loss for locally supervised training that alleviates the problem from greedy supervised learning, which collapsing task-relevant Information at the beginning of the layers.
The infoPro loss requires an auxiliary network to infer the Information I(h,y) (For ImageNet, the paper used ASPP).

Strong points:

Analysis of the phenomenon when applying the greedy supervised learning (GSL):
The two observations from the authors seem natural.

(1): GSL underperforms E2E training (little trivial for the reviewer)

(2): Each separated layer trained by GSL captures more discriminative features.

However, the information-based explanation of why the task-relevant info at the early stage of the network is essential (information collapse hypothesis) seems new and worth considering for the reviewer.

InfoPro-loss:
Theoretical analysis of the loss seems credible to the reviewer.

Experiments:
Mutual Information at each layer index looked interesting.

Interestingly, the proposed method outperforms E2E training in the ImageNet classification task.

Hyper-parameter does not seem to be sensitive.

Week point:

Using the additional network might not be fair when strictly comparing the performance. But practically, when training the large network, the size of the auxiliary network seems ok.

Comments:

(1) It is not strongly required. Is it available to apply the method to the detection task?

(2) Adding the experiments for other widely used network s.a. VGG, would be meaningful.

(3) Can we use LR-ASPP in MobileNet v3 instead of ASPP? or Can we use a heavier Decoder for better performance?

(4) How come when we set K=4 and set a larger batch size? We do not need to waste the remaining memory.

(5) Training the network larger than ResNet-152 from the proposed method would be impressive.

(6) Can we replace the imageNet pre-trained (by E2E) backbone with that trained by the proposed method? Experiments or discussion verifying the transferability of the trained network would be required.


Rating: Nice paper, Accept

(1) But the reviewer wants a discussion with the author about the comments mentioned above

(2) Literature checks from the other reviewers may affect the rating.

---

> ### Author Response · Authors · 2020-11-20
> **Response to Reviewer 1 (part 2)**
>
>
>
> **Q5**: How come when we set K=4 and set a larger batch size? We do not need to waste the remaining memory.
> **A5**: As a matter of fact, we show the effects of using larger batch sizes in semantic segmentation (Table 5). We also preliminarily test enlarging the batch size of InfoPro*, K=4 on CIFAR-10 from 1024 to 2048, but the improvement seems not significant (6.93 --> 6.86). However, we show in Table 5 that a more efficient approach to exploiting the memory saved by InfoPro* might be training models using data with higher-resolution. When increasing the crop size from 769x769 to 1024x1024, InfoPro* outperforms the end-to-end trained DeepLab-V3 by 0.65% in terms of mIOU. Note that this does not involve additional training or inference cost.
>
>
> **Q6**: Training the network larger than ResNet-152 from the proposed method would be impressive.
> **A6**: Thank you for the encouraging suggestion! However, due to the limitation at time, computational resources and the space of the paper, it might be difficult to report this result in the revised paper. We believe that the presented results with the large ResNet-152 and ResNeXt-101, 32×8d on ImageNet strongly verify the effectiveness of the proposed method. We will focus on the networks larger than ResNet-152 in our future work.
>
>
> **Q7**: Can we replace the imageNet pre-trained (by E2E) backbone with that trained by the proposed method? Experiments or discussion verifying the transferability of the trained network would be required.
> **A7**: Thank you for the suggestion. To verify the transferability of the models trained by the proposed method, we initialize the backbone of a Faster-RCNN using ResNet-101 trained by E2E training and InfoPro*, and train the network for the object detection task on MS COCO. The results are presented in Appendix I (due to the spatial limitation of the paper).

---

> ### Author Response · Authors · 2020-11-20
> **Response to Reviewer 1 (part 1)**
>
> Thank you very much for providing the valuable suggestions. Our responses to the comments are listed below:
>
>
>
> **Q1**: Using the additional network might not be fair when strictly comparing the performance. But practically, when training the large network, the size of the auxiliary network seems ok
> **A1**: Indeed, we involve additional networks. However, as you noted, the computational overhead is minimal. In addition, our new results, as shown in Figure 4, indicate that their size can be further reduced.
>
>
>
> **Q2**: It is not strongly required. Is it available to apply the method to the detection task?
> **A2**: (1) The formulation of the proposed InfoPro loss is general and flexible. As a matter of fact, its final optimization objective merely involves I(h, x) and I(h, y). We believe that, once these two terms are reasonably estimated, it can be applied to most of deep learning tasks, including object detection. For example, when dealing with image-based object detection tasks, we can estimate I(h, x) with the same decoder as image classification and semantic segmentation. The task-relevant information I(h, y) can be estimated with existing detection heads, such as FPN+RPN or the head used in YOLO. One may need to add a decoder together with a detection head at the end of each local module, attach a reconstruction loss to the regular detection loss function, and sequentially trigger the back-propagation process of all local modules with every mini-batch of training data.
>       (2) However, due to the limitation at time, computational resources and the space of the conference paper (and the CVPR ddl, frankly :D), we prefer to add the detection results to the extended journal version of this paper. We are now focusing on it. If we finish before the deadline of discussion, we will present the results in Appendix. Last but not least, we believe that the current results of image classification and semantic segmentation on five widely-used datasets might be sufficient to evaluate our method comprehensively.
>
>
>
> **Q3**: Adding the experiments for other widely used network s.a. VGG, would be meaningful.
> **A3**: We have added the results with VGG in Appendix I (due to the spatial limitation of the paper).
>
>
>
> **Q4**: Can we use LR-ASPP in MobileNet v3 instead of ASPP? or Can we use a heavier Decoder for better performance?
> **A4**: (1) At first, we hope to clarify a (possibly) potential misunderstanding. The ASPP head is used to estimate the task-relevant information I(h,y) for the semantic segmentation task on Cityscapes, instead of ImageNet. We introduce the architecture of the decoder and the auxiliary classifier used on ImageNet in Appendix E. Both of them follow a regular design, and involve little computational cost.
>       (2) The LA-ASPP module can be applied to InfoPro since it is also designed for obtaining the segmentation results. However, for a fair comparison with the end-to-end trained DeepLab-V3, we simply report the results with the basic version of ASPP.
>       (3) To investigate how the size of the decoder affects the performance of InfoPro, we scale its width with a certain factor (which equals 1 for the original design) and present the corresponding test errors. The results are shown in the left plot of Figure 4 in the revised paper. We find that the current decoder is generally sufficient for achieving the excellent performance. Using larger decoders merely improves the accuracy slightly.

---

### Decision · Program_Chairs · 2021-01-07
**Final Decision**

**Decision:**

Accept (Poster)

**Comment:**

All reviewers agree that the paper brings new knowledge in the field of locally supervised learning, and as such it should be accepted.  The authors should keep all reviewers' comments into account when preparing their camera ready version.